# Sedimentary blue carbon dynamics based on chronosequential observations in a tropical restored mangrove forest

Raghab Ray[1*], Rempei Suwa[2], Toshihiro Miyajima[1], Jeffrey Munar[3], Masaya Yoshikai[4], Maria Lourdes San Diego-McGlone[3], Kazuo Nadaoka[4]

[1]*Atmosphere and Ocean Research Institute, The University of Tokyo, Kashiwa*
raghab.ray@aori.u-tokyo.ac.jp; miyajima@aori.u-tokyo.ac.jp
[2] *Japan International Research Center for Agricultural Sciences, Tsukuba*
swrmp2005@yahoo.co.jp
[3] *Marine Science Institute, University of the Philippines, Diliman*
mlmcglone@msi.upd.edu.ph; jcmunar1@up.edu.ph
[4] *School of Environment and Society, Tokyo Institute of Technology, Tokyo*
yoshikai.m.aa@m.titech.ac.jp; nadaoka.k.aa@m.titech.ac.jp

*Correspondence to:* Raghab Ray (raghab.ray@aori.u-tokyo.ac.jp, raghab.ray@gmail.com)

**Abstract.** Among the many ecosystem services provided by mangroves, the sequestration of large amounts of organic carbon (OC) in marine ecosystems (also known as 'blue carbon') has given these unique ecological environments enormous global attention. While there are many studies on the blue carbon potential of intact mangroves (i.e., naturally growing), there have been very few studies on restored mangroves (i.e., planted). This study aims to address this knowledge gap by examining the sediment development process during the early colonization (rehabilitation) of mangroves in an OC-poor estuary in Panay Island, Philippines. Based on source apportionment of multiple endmembers in the sedimentary organic matter, the contribution of mangrove plant material was higher at the older sites compared to the younger sites or bare sediments where there is more contribution of riverine input. A clear increasing gradient according to mangrove development was observed for bulk OC (0.06–3.4 µmol g$^{-1}$), porewater OC (292–2150 µmol L$^{-1}$), sedimentary OC stocks (3.13–77.4 Mg C ha$^{-1}$), and OC loading per surface area (7–223 µmol m$^{-2}$). The estimated carbon accumulation rates (6–33 mol m$^{-2}$ yr$^{-1}$) based on chronosequence are within the global ranges and show an increasing pattern with the age of mangroves. Hence, the sediments of relatively young mangrove forests appear to be a significant potential C sink, and short-term chronosequence-based observations can efficiently define the importance of mangrove restoration programs as a potential carbon sequestration pathway.

## 1. Introduction

The term 'Blue Carbon' was first introduced more than a decade ago to describe the large quantity of organic carbon (OC) present in shallow coastal habitats like mangroves (Nellemann et al., 2009). Mangroves located around tropical and subtropical coastal regions are known for storing significant amounts of OC in the sediment and vegetation biomass. Out of the typical total carbon stocks of 739±28 Mg C ha$^{-1}$ in mangrove ecosystems, sediment OC accounts for 73–79% (down to 1-m depth) while above- (AGB) and below-ground root biomass (BGB) account for 14–15%

and 8–9%, respectively (Alongi, 2020; Hatje et al., 2020; Walcker et al., 2018). Mangrove sediment is the largest depository of OC owing to their efficiency in trapping suspended sediments and associated sedimentary organic matter (SOM), high algal OM (benthic algae and phytoplankton) and vascular plant-derived OM and low decomposition rates of SOM under anoxic conditions in the sediment. A 0.5- to 3-m-depth core generally represents ~49 to 98% of

ecosystem OC stocks (Donato et al., 2011), and sediment depth in undisturbed mangrove forest sites can often exceed 3 m (Elwin et al., 2019). High C sink capacity of mangroves makes them one of the key ecosystems that contribute to climate change mitigation by capturing large amounts of atmospheric $CO_2$ (Howard et al., 2017). However, there has been a rapid loss of mangroves due to land use and deforestation that has resulted in the release of OC stored in the sediments back into the atmosphere as $CO_2$ (Valiela et al., 2001). For instance, the conversion of mangroves to

aquaculture ponds, paddy fields, and pastures, and the removal of mangrove trees have resulted in OC stocks per unit area becoming 1/8–1/2 of the intact mangrove forests (Salmo et al., 2013; Kauffman et al., 2017; Sharma et al., 2020). Therefore, the quantification of blue C stocks and sequestration rates provides added value to mangrove protection as an ecosystem service and serves as a useful management tool when implementing plans for mangrove sustainability and productivity (Sheehan et al., 2019). During the last three decades, several countries have implemented mangrove

rehabilitation and restoration programs effectively to reverse mangrove forest cover loss. However, mangrove restoration efforts such as Reducing Emissions from Deforestation and forest Degradation (REDD+) have considered C emission loss only from above-ground biomass (Pendleton et al., 2012). Countries like the Philippines have started to address REDD+, loss of mangroves, and degradation of blue C habitats through their policies and rehabilitation management plans. To assess the effectiveness of such efforts, there should be a comparison between the intact and

restored mangrove forests in terms of sediment OC stocks and accumulation rates. Restored mangrove forests are rarely explored globally with some notable exceptions in the subtropical coastal regions of China (Ren et al., 2010; Lunstrum and Chen, 2014; Wang et al., 2021), and Vietnam (Van Hieu et al., 2017; Dung et al., 2016). In this study, an evaluation based on a type of 'natural experiment' or chronosequence (a.k.a. "space-for-time-substitution" or SFT; Pickett, 1989) was conducted at a relatively younger site (e.g., a mangrove ecopark in the Philippines) wherein, to

fulfill the conditions for chronosequence, all environmental and biological conditions of the experimental sites must be identical except for the age, and the species diversity is low (Nilsson and Wilson, 1991; Walker et al., 2010). The judicious use of chronosequential observation or SFT has already advanced our understanding of short-term temporal dynamics of carbon in naturally expanding mangroves (e.g., 66-year extent in Walcker et al., 2018; 70-year extent in Kelleway et al., 2016).

The capacity of nearshore vegetated habitats as blue carbon sink is controlled by geophysical constraints such as sediment supply rate, depositional conditions, and tidal elevation (Miyajima et al., 2017; Jiménez-Arias et al., 2020). Based on chronosequential studies of naturally occurring mangroves, OC accumulation in sediments increase with tree age, and OC sources change spatially with mangrove development (Lovelock et al., 2010; Marchand et al., 2017; Walcker et al., 2018). Most of the OC stored in mangrove sediments change at the spatial scale from plant-derived

OM at the interior mangrove sites to algal OM at the proximal tidal flat (Gontharet et al., 2014; Prasad et al., 2017; Ray et al., 2018). With the development of mangroves, higher vascular plant or mangrove-derived OC sources may dominate the OM pool (Marchand et al., 2006). A significant fraction of the mangrove-derived OC that has

accumulated on top of the bare sediment can be washed away to the nearshore waters by tidal action (Brown et al., 2021; Ray et al., 2020). By considering bare sediments and old growth mangrove stands as two extreme ends of a

transect that consists of mangroves with different ages, a systematic overview of the sedimentary blue C dynamics can be captured for restored mangroves. The stable isotope ratio of carbon ($\delta^{13}$C) is frequently used to evaluate the relative contributions of endmember sources to the OM pool through mixing models with either $\delta^{15}$N or C:N ratios (Ray and Shahraki 2016; Sasmito et al., 2020). The use of these biogeochemical controls on blue C dynamics has rarely been reported for restored mangroves (e.g., *Kandelia*-dominated forest reported by Van Hiew et al., 2017).

In this study, we address the question of how chronosequential observations in a restored mangrove forest could serve as guide in achieving an improved scientific understanding of C sources and stocks and monitor the changes in accumulation rates in the early development stage and adult stages. Here, we hypothesize that restored mangroves increase sediment C storage in accordance with the maturity of the vegetation. To test this, we (1) calculated the total OC (TOC), dissolved OC (DOC), and OC accumulation rate along a chronosequence of restored mangroves forests

located in the Philippines, and (2) examined how blue C varies with sedimentary geochemical properties (OC, bulk density, specific surface area). Isotopic signatures such as $\delta^{13}$C, which allows for an efficient provenance analysis of SOM, were also examined. Additionally, particulate OC (POC) present in the surface water was analyzed to assign different endmember sources in the SOM pool (e.g., plant organ, riverine and pelagic algae).

**2. Material and Methods**

**2.1. Study area**

Sampling was conducted in a planted mangrove forest, locally known as Bakhawan Ecopark, located in Kalibo City in Panay Island, central Philippines during the wet season (September 2018 and 2019) and dry season (February 2019) (Fig. 1, 11° 43' N, 122° 23' E). The Bakhawan Ecopark is the remnant area of a former deltaic mangrove at the mouth of the Aklan River (Duncan et al., 2016). Aklan River, which has a drainage area of 852 km$^2$, flows into the

northwestern coastal area of Kalibo, continuously depositing sediment to form the alluvial plain down the river. Sediments entrained by the longshore current formed sandbars, beach ridges, and coalesced mouthbar deposits. To prevent the damages by coastal flooding, a large portion of the sea-facing mudflat was planted with 45 ha of *Rhizophora apiculata* and 5 ha of *Nypa fruticans* in 1990 by a cooperative comprised of local families (Kalibo Save the Mangroves Association or KASAMA). An additional 20 ha of *Rhizophora* spp. were planted in 1993 (Primavera,

2004) for the purpose of stabilizing the shoreline, decreasing sedimentation offshore, and increasing fish stocks and wood production (Department of Environment and Natural Resources or DENR, Philippines). Insect damage to the plantation in 1997 was followed by infilling of naturally recruited *Avicennia marina* and *Sonneratia alba*. The seafront area was replanted in 2006 with *Rhizophora apiculata*, and subsequently recolonized naturally by *A. marina and Sonneratia alba* (Duncan et al., 2016). New recruitment of both *A. marina* and *R. apiculata* took place on the mud

bank in May/June 2019. The inland part of the Ecopark is dominated by naturally growing mangroves. The natural growth of mangrove trees and planting efforts since the 1990s at the Aklan River mouth stabilized and enlarged the mangrove forest by at least 627% to a flourishing 121 ha today. Based on remotely-sensed data, it was found that the land area of the forest increased by 52.4% on average every five years since 1985 (Landicho et al., 2018). The Food

and Agriculture Organization of the United Nations has cited the Bakhawan mangroves for excellence in forest management (Cadaweng & Aguirre 2005).

The tide in the Bakhawan Ecopark is semidiurnal microtidal with the highest amplitude of around 2m. The mangrove forest floor is fully inundated during high tide. At the mouth of Aklan River, water meanders along a small channel between the sandbar and mangrove-lined coast. The climate of Aklan is categorized as Type III (according to the Philippines Atmospheric, Geo-physical, and Astronomical Services Administration) with no pronounced maximum rain period except for short dry periods of 1–3 months (December to February or March to May). The rest of the year represents the wet season with a total annual rainfall of 3200±775 mm and a mean temperature of 27.2 °C (2017–2018, JRA-55 reanalysis).

### 2.2. Mangrove chronosequence

Sediment sampling locations are different from each other in terms of mangrove development, elevation from mean sea level, and inundation pattern. Sediment sampling locations were categorized according to mangrove age; these are bare sediments (BS, 0-yr), pioneer mangroves (PM, 0.25-yr), young mangroves (YM, 10-yr), adult mangroves (AM, 20-yr) and mature mangroves (MM, 30-yr). The ages of the mangroves are typically known from their plantation period (Salmo et al., 2014). In this study, mangrove categories are partly influenced by Fromard et al. (1998) who examined the chronosequential sedimentary OC in naturally growing *Avicennia*-dominated mangroves in the French Guiana muddy coast where PM were established on the seafront after stabilization of mud banks, or on the sandy offshore bar (height <2 m), followed by further maturation to younger stands (YM, height <8m). According to Fromard et al. (1998), both PM and YM colonize rather unstable marine clays/sands that are regularly flooded by tides. From the river mouth to upstream, the stands (adult and mature) become older and taller (8–15 m, *Rhizophora spp.* in French Guiana), phenomena that are linked to river dynamics rather than tidal movement. In Bakhawan Ecopark, MM and AM sampling sites are farthest away from the water areas while BS and PM are closest to the sea (Fig. 1). The center of the mangrove forest is dominated by AM and the sea-facing edge of the Ecopark has decreasing mangrove age from MM to PM. Both BS and PM are completely inundated during the high tide while MM and AM are partly inundated. There is a steep increment in elevation from seaward to landward sampling sites (–1.2 to 0.45 m; refer to section 3). Seaward sites are characterized by sandy sediments compared to silty/clay sediments at the landward sites. *Rhizophora apiculata* is the dominant species at YM, AM and MM, while mixed mangroves (*Avicennia* and *Rhizophora* sp.) compose the PM. Between the two sites of bare sediments, BS1 which is closer to YM, was sampled during the wet season, while BS2 which is isolated from the mangrove sites, was sampled in the dry season, (Fig. 1). Mean tree heights were <1 m, 3–4 m, 6–8 m, 10–15 m for PM, YM, AM, and MM, respectively (data not shown).

### 2.3. Sampling procedure

The variables tested at each sampling site from BS to MM were sediment thickness, coarse fraction, pH, ORP, bulk density, specific surface area (SSA), concentrations and isotope ratios of carbon and nitrogen, and pore water dissolved organic carbon (DOC). Sediment thickness was measured at each site using a tool for cone penetration test (KS-159, Kansaikiki Inc.) (Yoshikai et al., 2021). Single cores were collected at each site during low tide by manually pushing

an Eijkelkamp peat sampler (DIK-105A, 52-mm Ø, 50-cm length) into the sediment. A total of 8 cores were retrieved during the survey period with seasonal collections obtained from BS, AM, and MM sites (dry and wet seasons, a total of 6 cores) and from PM and YM sites (wet season, one core each). The GPS coordinates of all sampling sites were recorded (Garmin MAP64s) to locate them again for duplicate sampling in a different season. Immediately after sampling, each core was sectioned into 2-cm intervals up to the first 10 cm and 5-cm intervals beyond 10-cm depth. The total sample depths at BS, AM, and MM were always 50 cm, while for PM and YM were 20 cm and 25 cm, respectively. *In situ* pH (NBS scale), temperature and oxidation reduction potential (ORP, Pt-electrode) were recorded for each section using hand-held multiparameter probes (HORIBA pH-conductivity sensors, WTW redox sensor). About 0.8–1 kg cores were collected from each site. Visible root material, decaying plant matter, and dead wood were removed from the sediment sample in the field. Within 3–4 hours after collection, the sediment samples were kept in a styrofoam box and brought to the laboratory for analysis of bulk density, SSA, concentrations and isotope ratios of carbon and nitrogen.

Additional sediment cores for porewater sampling were collected by inserting a PVC corer manually at each site (6.5-cm inner diameter, 70-cm length). A total of 7 cores were retrieved during the study period (single core from each site per season). Immediately after retrieval, the top and bottom ends of the corer were closed using rubber caps. Rhizon tubes (Rhizosphere Research Products) were inserted to the holes drilled at specified intervals (0.5-cm interval until the first 4 cm followed by 2-cm interval up to 10 cm, and finally 5 cm interval up to the deepest layer). A Rhizon tube is a small microporous polymer tube (2.5-mm diameter, < 0.2 µm pore size of the membrane) connected to a plastic syringe (25-ml capacity) by a standard Luer-lock connector. Around 8–10 ml of porewater were extracted in about an hour. The salinity of the extracted porewater was measured using a refractometer. Porewater was transferred to pre-combusted amber vials (20 mL) for DOC analysis.

Tidal water sampling was conducted along the salinity gradient (0 to 33) in the Aklan River. For this study, three representative sites were chosen, the upstream of the Aklan River and outer shore as potential endmember sources of SOM, and the river channel very close to BS site during high tide. More details of water sampling techniques can be found in the supplementary material. A global tide prediction model (NAO.99b, Matsumoto et al., 2000) was used to correct water depth data to the relative elevation at each site from the Mean Sea Level or MSL.

In the laboratory, pre-weighed wet sediment subsamples were oven dried at 60°C for 48 h to allow for the calculation of dry bulk density (BD) and water content. For sediment subsampling, an open mouth plastic syringe was used (2-cm inner diameter and 1.5-cm length). Bulk density (g cm$^{-3}$) was determined as the dry sediment weight (g) divided by the initial volume (cm$^3$). The rest of the wet sediment samples were freeze-dried using a Benchtop Freeze Dry System (Labconco). The freeze-dried samples were gently crushed using a mortar and pestle and passed through a 1-mm mesh stainless steel sieve to remove large gravel (referred to as the coarse fraction). The sieved samples were stored in tightly capped glass vials under <40% relative humidity.

**2.4 Chemical analyses**

The dried and homogenized sediment samples were subjected to acid treatment to remove inorganic carbon. Approximately 1 g of dried sample was placed into screw-capped glass tubes (10 mL), and 2.0 N hydrochloric acid (HCl) solution was added dropwise until all the carbonate was converted to $CO_2$. After centrifugation for 15 min at 2000 rpm (G-force 760 g), followed by washing with deionized water and decantation, the final residue in the tube was dried at 60 °C overnight. Once cooled, the dried samples were weighed into a tin capsule (10 x 10 mm), folded, and kept temporarily in a 48-well microtiter plate until analysis.

The concentrations and isotope ratios of OC and total nitrogen (TN) in the treated samples were determined simultaneously by EA-IRMS (FLASH 2000/Conflo IV/DELTA V Advantage, ThermoFisher Scientific, Bremen, Germany). Two standard materials of different $\delta^{13}C$ (–26.4 to –19.6‰) and $\delta^{15}N$ (2.5 to 5‰) values (SI Science Ltd., Saitama, Japan) were used for calibration. The measured isotope ratios were presented using the conventional δ-notation ($\delta^{13}C$ and $\delta^{15}N$, in ‰) with Vienna Pee-Dee Belemnite and atmospheric $N_2$ as the reference materials. The instrumental analytical precision was normally within ±1% for the OC and TN concentrations and ±0.1‰ for $\delta^{13}C$ and $\delta^{15}N$.

Measurement of SSA of the sediment samples was performed by the multipoint Brunauer–Emmett–Teller (BET) method based on $N_2$ gas adsorption under reduced pressure (for details refer to Miyajima et al., 2017). Specific surface area was measured only for BS, AM, and MM cores. The dried and homogenized sediment samples were heated at 350 °C for 12 h followed by calculation of weight loss on heating after 1 hour. Between 0.5 g and 2.0 g of the treated samples were weighed into glass flasks and desiccated in vacuo at 350 °C for 3 h. Immediately after cooling, a multipoint BET measurement was performed with $N_2$ (purity > 99.99%) as the adsorbate using a BELSORP mini II (MicrotracBEL, Osaka, Japan) surface area analyzer. The slope of the BET plot in the linear region was used for estimating the SSA.

**2.5 Data analyses**

The results of the OC and TN concentrations, and the SSA were expressed as µmol and $m^2$ per unit dry weight of bulk sediment, respectively. Carbon stocks (Mg C $ha^{-1}$, top 50 cm, referred to as $S$) of each core at the 5 sites were calculated as

$$S = \sum_{i=0}^{n} Cn \times \rho n \times l \dots\dots\dots\dots\dots\dots\dots\dots\dots\dots\dots\dots\dots\dots\dots\dots.(1)$$

where $Cn$ is the C concentration (mass %) and $\rho n$ is the bulk density (g $cm^{-3}$) of the sample, and $l$ (cm) is the length of the sample section (i as depth zero to $n^{th}$). The amount of OC preserved per unit surface area of sediment particles is referred to as OC loading (OC/SSA in µmol $m^{-2}$; Mayer 1994).

In a chronosequence study, it is common to apply linear regression model between mangrove age and C stocks and derive the slope, i.e., carbon accumulation rate (CAR) (Alongi, 2004; Walcker et al., 2018), or individually following the formula such as CAR = [(C stock of the stand) - (C stock of the previous stand)] / [(age of the stand) - (age of the previous stand)] (Marchand et al., 2017). However, our results showed best-fit with the exponential function for the relationship:

$$OC\ stock\ (mol/m2) = a.\,e^{b.yr} \dots\dots\dots\dots\dots\dots\dots\dots\dots\dots(2)$$

where *a* and *b* are the constants determined from the best-fit exponential relationship, and *yr* is the age of mangrove stands (years). Here we assume that with the exponential increase of plant above-ground mass with early mangrove growth, below-ground root mass also increases, which contributes significantly to OC accumulation with early mangrove development. However, because the exact maturity stage of these mangroves is unknown, the typical logistic curve equation that is otherwise used for matured forests was not applied, instead the best-fitted exponential trend was used. Based on the exponential model, the slope was derived for the individual sample following the equation:

$$Slope\ or\ CAR\ \left(\frac{mol}{m2.yr}\right) = \left(\frac{dOC}{dt}\right) = a.\,b.\,e^{b.yr} \dots\dots\dots\dots\dots\dots\dots(3)$$

Accordingly, CAR for the age class of mangroves (0.25, 5, 10, 15, 20, 25 and 30 year) was calculated and fitted with an exponential function following a non-linear least square method using *nls* function of the statistical software R ver. 4.0.2 (R development core team, 2020). The 95% confidence interval for the fitted model was computed with *predFit* function of *investr* package. The exponential model was compared with a simple linear regression model.

To evaluate the effects of sampling Depth and stand Types (i.e., chronology based) on TOC, porewater DOC, $\delta^{13}$C, $\delta^{15}$N and OC:TN ratio of SOM, the general additive linear mixed model (GAM) was constructed with a *gam* function in a R package (mgcv). Results from subsamples of each core were used to for the model (total core = 8, subsamples used for analysis = 92). Using the results of *gam*, ANOVA was used to evaluate the significance of each variable with a significance level at $p = 0.05$.

For subsequent provenance analysis, a binary source mixing model was applied (Parnell and Inger, 2016) based on the mean values of $\delta^{13}$C, $\delta^{15}$N, and C:N ratio of sediment samples from different depths in each core. The samples obtained in different seasons were treated as independent samples. In total, seven mean values were used for the analysis of Bayesian mix model using multiple R packages (RGtk2, splanc). The endmember sources of OM chosen for this study were the green leaf of *Rhizophora apiculata*, POM (particulate organic matter) of marine and river water, and microphytobenthos (MPB). Among these endmembers except for POM, the $\delta^{13}$C, $\delta^{15}$N, and OC:TN ratio for green leaf and MPB were obtained from literature (refer to Table S1). Although OC:TN values were shown to be largely variable between green (~30) and yellow/senescent leaves on the sediment floor (~50) of *Rhizophora apiculata*, their $\delta^{13}$C values were the same (–28.5‰) (Nordhaus et al., 2017). For this similar reason the mangrove root was also not considered; instead, the leaf singularly represented the mangrove-derived OC source. Hence, it is reasonable for this study to choose green leaves and avoid redundant increase of the number of endmembers for the model. Microphytobenthos located on the tidal flat between the interface of sea and land can also be variable depending on the $\delta^{13}$C of dissolved inorganic carbon (photosynthesis substrate for MPB). This is similar to the riverine settings in French Guiana ('mixed MPB' –20.9‰, Ray et al., 2018), in contrast to the oceanic setting in the Red Sea ('pure MPB' –17.9‰, Shahraki et al., 2014). The number of endmembers were chosen carefully by keeping the small standard deviation of $\delta^{13}$C and OC:TN within each endmember and making sure the values are clearly separated with significant differences from each other (ANOVA, p<0.05). We refrained from considering a large number of endmembers and

omitted less important endmembers or treated some endmembers with similar isotopic and elemental compositions (e.g., plant parts such as leaf, root, litter) collectively. Since there was no seagrass recorded at or around the sites, its potential endmember contribution to the mangrove SOM was ignored. On the contrary, although the isotopic composition of MPB and marine POM were not very different, we selected MPB as endmember due to its visible presence on the exposed tidal flat (BS2) during the low tide.

## 3. Results

Mangrove development with age is shown in Fig. 2a. Maximum elevation of 0.45 m from the MSL was measured at the MM, followed by AM and YM (0.35), PM (0.2) and BS (−1.3). Mean porewater salinity, and pH did not vary significantly between the various stages of mangrove development, but sediment temperature showed higher values at BS than the vegetated sites (Figs. 2b-d). ORP were recorded maximum at older sites (Fig. 2e). Sediment thickness was higher in mangrove sediments (140–265 cm) than the tidal flat (<100 cm) (Fig. 2f). The fraction of coarse sand (>1 mm mesh size) tend to be higher seaward (>6% at BS, Fig. 2g). Sediment bulk density at the top 50 cm increased with decreasing mangrove age tending to be maximum at BS (0.6 to 1.3 g cm$^{-3}$, Fig. 2h). Specific surface area of the sediments varied from 9.8 to 21.2 m$^2$ g$^{-1}$ ((Fig. 2i, n = 19) with the lowest value for the coarser bare sediment. A consistent increment in sediment OC concentration, TOC:TN molar ratio (from YM to MM), and porewater DOC was observed with mangrove development (Figs. 2j-l). The mean value of bulk $\delta^{13}$C became more negative (Fig. 2m), and the mean $\delta^{15}$N became more positive with mangrove age ($\delta^{15}$N = −1.15 at BS to 1.06 ‰ at AM, figure not shown). Organic carbon (OC) stock was determined to be maximum at the mature stand (93.5 Mg C/ha) and lowest at the bare sediment (3.13 Mg C/ha) (Fig. 2n). Similar to the trend of OC stocks, OC loading varied widely between 4 and 380 μmol/m$^2$ (Fig. 2o), with mangrove sites having higher OC loading than BS.

Vertical profile of sediment OC and $\delta^{13}$C, specific surface area (SSA) and OC loading, and porewater DOC on a seasonal basis are shown in Figs. 3–5. Sediment surface values (0–10 cm) of porewater salinity, pH and ORP changed significantly with mangrove ages in both seasons (p < 0.05, Fig. S1), except for porewater salinity at the bare sediment during the dry season. The deeper layers of each sediment core (>10 cm) did not exhibit much variability in their properties. The minimum pH was recorded at the top 10-cm depth of the mature mangroves in the wet season (5.41±0.26). Vertical profiles of TOC and $\delta^{13}$C showed wide variations than the physico-chemical properties (Figs. 3a–d). Notable peaks for TOC were observed in mature mangroves and adult mangroves (around 20-25 cm). Total organic carbon and $\delta^{13}$C in bare sediment changed very little with depth, especially in the dry season. Significant differences in OC/TN molar ratios among the sediments from different depths of MM and other sites were observed (Figs. 3e,f). At the non-vegetated sites (BS), the ratios varied slightly with seasonal changes. The vertical profile of SSA based on the values at three specific depths of each core showed a mild decreasing trend from the surface to the deepest layer (50 cm) in both dry and wet seasons (Figs. 4a,b). The depth-specific values of SSA were always higher at the adult mangrove sites than the other two sites (except at 45–50 cm in the dry season). Similar to SSA, OC loading also decreased with core depth, and mature mangroves exhibited maximum loading without seasonality (Figs. 4c,d). All stages of mangrove development showed an overall increase of DOC concentration with depth regardless of the seasons (Figs. 5a,b). The DOC concentrations of deeper layers reached four to five times higher than those of the

upper sediment in the wet season, and two to three times higher in the dry season. For example, our data demonstrated a change from 600 to around 4000 µmol L$^{-1}$ DOC concentrations for the adult mangrove in both the dry and wet season. A wide range in DOC of 128 to 920 µmol L$^{-1}$ was also measured for the non-vegetated bare sediment site.

Using the generalized additive model (GAM) as the basis, the results of ANOVA for the main parameters showed significant dependence of TOC over mangrove types (F = 42.88, p<0.001), and depth (F = 4.11, p<0.001) (Table 1). Porewater DOC varied significantly with mangrove types (F = 4.92, p<0.005), similar to δ$^{13}$C (F = 29.9, p<0.001) and δ$^{15}$N (F = 4.02, p<0.05). The changes with depth as smooth term for TOC, DOC and δ$^{13}$C based on the GAM are given in Fig. S2.

The OC/TN and δ$^{13}$C of the five mangrove types were plotted with the four endmember sources (River POM, Marine POM, Leaf and MPB) to rectify the sources of SOM ( Fig. 6), and subsequently calculate the endmember contribution to each type ( Table 2, Fig. S3). Except for MPB and marine POM, the relative contributions of mangrove leaf and river POM were significantly different for each site (one-way ANOVA, p-value mangrove leaf <0.05, p-value river POM <0.05). At the bare sediments and pioneer mangrove sites, river POM dominated the SOM pool with mean contributions of 58% and 43%, respectively, whereas at the young and mature stands, mangrove leaf was the main potential contributor (53% and 58%, respectively). However, SOM remained more as a mixture of river POM and leaf material (~40%) at the adult mangrove sites. The contributions of benthic and pelagic algae as SOM sources were not very significant for the entire mangrove sediment in the area studied.

In the water part, results from three locations (upstream-channel-offshore) showed salinity changing from 0 to 33, pH increasing from 7.6 to 8.1, and DO (%) increasing from 89% to 105% to downstream (Table S2). Surface water POC was 4–5 fold lower than the DOC in the upstream and channel water (~ 20 µmol L$^{-1}$ versus 90 µmol L$^{-1}$), and lowest in offshore water (10 µmol L$^{-1}$). The δ$^{13}$C-POC was highest in offshore water (–22.8‰) and lowest upstream (–25.9‰).

### 4. Discussion

#### 4.1 Relevance of chronosequence approach

Several studies have relied on direct measurement of carbon accumulation rates (CAR) by combining sedimentary C content and soil accumulation rates estimated from radioisotopes, $^{210}$Pb and $^{137}$Cs or natural markers like volcanic ashes (Sanders et al., 2010). These CAR estimations are based on the assumption that sediment and OC accumulation are in a steady state during the period of accumulation/deposition. However, the "indirect way" or the chronosequential theory assumes that the temporal variations in soil properties in different-aged sites fall into the same time trajectory of OC accumulation. This assumption requires a condition that these different sites had experienced similar driving factors of OC accumulation processes following tree growth after the restoration. While hydrological processes such as hydroperiod and the tidal regime are considered important drivers of OC accumulation, it is regarded that these conditions do not vary largely among the sites and have been relatively stable over the time window of concern (~30 years after restoration) given the same level of ground elevations in the same forest. Also, given the significant fraction of mangrove contribution to SOC (Table 2), the influence of C inputs from external systems may not be a significant

factor in shaping the OC accumulation trajectories in the sites. Only the vegetation structures vary significantly among sites, which may characterize the evolution of OC stocks with forest age in our study site, the Bakhawan Ecopark (Fig. 7). In addition, we consider that the AGB and BGB development in the sites follow a similar trajectory given the same level of soil salinity (Fig. 2) – one of the most important regulators of mangrove growth – and the same plantation spacing and species. Therefore, it is reasonable to consider that the OC accumulations in the different-aged sites also follow a similar time trajectory, and thus Eq. (3) can be applied to estimate CAR in our study site.

## 4.2 Sediment condition and organic matter

The results of dry bulk densities and granulometry (coarse fraction, SSA) of the tidal flat and mangroves in Bakhawan Ecopark indicate that soils are relatively homogeneous with fine-grained fractions relatively prevalent towards mangroves at higher elevation (mainly towards AM), while coarse-grained sands are more common on the tidal flat and younger mangroves (BS and YM, PM) towards the shore (Fig. 2). Bulk densities (BD) at the sampling sites (0.3–1.3 g cm$^{-3}$) were comparable with the reported BD values across mangrove soils of the Indo-Pacific regions (0.20–0.92 g cm$^{-3}$; Donato et al., 2011), with sand fractions dominating the lower intertidal zone. Specific surface area is primarily constrained by grain size and its vertical profile is presumably related to the finer upward trend of sediment grain (Fig. 4a, c). The latter might have been a result of the sediment-stabilizing function of mangroves or influenced by weathering process that transforms sand and silt fractions into clay fractions (Shen et al., 2020). Furthermore, carbonate-bearing minerals can influence SSA more on the tidal flat than the organic-rich mangrove sediments. Among the older stands (MM and AM), higher SSA and sediment thickness, and low BD at the adult stands suggest the dominance of finer-grained clay material probably because of the long-term deposition of the weathered minerals, and the narrow sandbar halfway down the Ecopark (Fig. 1) that may have closed off the older mangroves from higher wave energy and shifted the deposits of finer grains. Total organic carbon is higher at the older sites than the younger ones due to fine-grained (silt + clay) sediments that tend to have higher TOC than coarse sands (Canfield, 1994). Fine-grained silt has larger SSA that creates higher capacity to adsorb OM (Mayer, 1994).

Among the physicochemical properties, lower porewater salinity at the BS than the mangrove sites indicate greater dilution from the direct input of river water, which becomes less with increasing elevation level inside the forest floor (Fig. 2). On a temporal scale, the overall low porewater salinity during the wet season is more likely linked to rainwater dilution (350–450 mm in September, 75–150 mm in February; JRA-55 Reanalysis) and elevated groundwater level. Because of the shading effect, surface sediment temperatures at the mangrove-vegetated sites were lower than the sun-exposed BS or PM. The lowest recorded ORP at BS was due to reducing conditions that prevailed through tidal water saturation.

It has been observed that mangrove sediments in the river-dominated estuary accumulate larger proportion of the mangrove-derived OM than those in the tide-dominated estuaries/oceanic mangroves dominated by marine algae (e.g., Indonesian mangroves, Kusumaningtyas et al., 2019; Latin American, Gontharet et al., 2014; Middle-East, Ray and Shahraki, 2016). In the microtidal riverine setting of the Bakhawan Ecopark, OM input from land sources is more dominant than marine sources at the mangrove sites (Fig. 6). Endmember mixing model suggests that there is a clear gradient with respect to the relative proportion of OM sources along the tidal flat-mangrove continuum (Fig. S3).

Sediments from cores at the bare sediments and pioneering mangrove sites show the predominance of OM probably derived from fluvial transport of eroded organic material within the catchment. Upon transport into the coastal area, fine sediment and POM accumulate at the calm shallow water channel of the topographically lowest elevation zone, and consequently, the longer inundation period facilitates sedimentation and deposition of the suspended matter. Similar values of $\delta^{13}C$ in the surface water POC upstream (–25.9‰, Table S2) and sediment OC at the bare sediments

and pioneering mangroves (mean –26‰) support their upstream allochthonous sources. A sizeable fraction of OC at the tidal flat was contributed by marine and benthic algae. In a companion study, other than the visible evidence of green algal patches, the measured day-time $CO_2$ uptake flux on the tidal flat contrasts with the emission flux at the forested sites (Influx at tidal flat: $-4$ to $-2$ mmol m$^{-2}$ h$^{-1}$, efflux at mangrove: $5-12$ mmol m$^{-2}$ h$^{-1}$, unpublished data). The strikingly low OC:TN ratio at the surface sediments and the minimum $\delta^{15}N$ suggests the presence of N$_2$ fixing

bacteria on the tidal flat during exposed tide conditions. The slightly higher mangrove contribution to SOM in pioneering mangroves (up to ~54%) clearly indicates additional OM input from the small growing plants along the channel.

At the topographically higher mangrove sites, greater contributions of autochthonous sources (i.e., mangrove plant materials) are correlated with lower $\delta^{13}C$ and higher OC/TN (>12). This is favored by the relatively high bed elevation

and decrease in submersion time, promoting the retention of detrital OM on the sediment layer (shown as leaf OM, Fig. S3). Although such evidence of greater plant input to SOM at interior mangrove sites compared to mudflats is not new for intact forests (Marchand et al., 2003; Sanders et al., 2010; Matos et al., 2020), this is rare for restored mangroves considering an extended gradient from mudflat to mangrove appearance (except in Vietnam; Van Hieu et al., 2017).

Terrestrial C3 plants, like mangrove plant organs, have C/N ratios of around 12 or higher (Prahl et al., 1980) and are N-poor due to the dominance of lignin and cellulose type of compounds. The significant positive correlation between TOC and TN ($r^2 = 0.96$ at BS, PM, YM, and AM; $r^2 = 0.42$ at MM, figure not shown) in the sediments indicates that the C and N in the samples are predominantly associated with the organic pool. It is noteworthy that such correlation is relatively poor at the mature stands. Although the exact reason for this is unknown, the abundance of benthic animals

during another complimentary experiment (burrow density at MM: 150±55, YM: 70±62, BS: 5±2 individual m$^{-2}$, Ray et al., unpublished) might suggest intense bioturbation and sediment remobilization.

### 4.3 Vertical profile of organic carbon

The variation in OC content from the bottom to the top of sediment cores also reflected the change in the proportion of mangrove-derived materials and allochthonous (e.g., benthic algae) organic carbon sources in the sediments, and

the mineralization of organic matter in the mangrove sediments (Tue at al., 2011). The difference in the sediment physicochemical and OC profile at the upper 50 cm of the layers may result from multiple factors that promote OM decomposition, such as immediate exposure of litter in the surface layers (e.g., managed *Rhizophora* in Malaysia, Ashton et al., 1999), while coarse and fine roots contribute to carbon and nutrients at shallower depths (reported down to 52 cm for 27-yr planted *Rhizophora* in Vietnam; Arnaud et al., 2021). Multiple mid-layer peaks of TOC are

sometimes observed which presumably reflects the influence of root biomass (Fig. 3). This study showed that root

activity within the sediment column is essentially dominant when comparing vegetated sites with bare sediments. Assuming a mixture of algae (marine POM plus MPB, $\delta^{13}C = -21.1‰$) and mangrove root/plant organ ($\delta^{13}C = -28‰$) as two major endmember sources of the sediment OM pool, provenance analysis confirms the maximum contribution of roots at the old stands (75–100%) and minimum contribution at BS and PM (52–72%), corroborating the TOC peaks observed at different depths (particularly between 10–25cm, Fig. 3a, b). It has been shown in other studies that root-derived carbon tends to be accumulated more efficiently as aggregates in the SOM pool and contribute largely as a potential C source (Xia et al., 2015).

Depth-wise patterns of OC were not much variable in both seasons (Figs. 3a, b), suggesting a dominant OM source prevailing down the cores. Root exudates like sugars and amino acids are suggested to be the main carbon source for the localized microbes (Bouillon and Boschker 2006; Jiang et al., 2017). As mentioned in the previous section, bioturbation might play an important role in carbon accumulation at the shallow sediment depth of the older stands where maximum burrow densities were observed compared to the tidal flats and younger mangrove sites. The role of burrowing crabs as a carbon sink has also been reported at deeper soil layers of *Avicennia* stands in Kenya (40–80 cm, Andreetta et al., 2013) where burrowing sesarmids provide a continuous supply of fresh organic matter down the profile. At low tide, the presence of water with low oxygen saturation, covering the bottom of the burrows avoids oxidation by creating an extension of the sediment-air interface and favoring greater OC accumulation (Kristensen, 2004; Smith III et al., 1991). The same study in Kenya reported the greatest OC concentration where the crab population was maximum, otherwise hypoxic or anoxic, and creating an extension of the coastal marshes sediment-air interface and favoring greater OC accumulation.

The depth-wise relationships between sediment OC and porewater DOC are less obvious (Fig. 3, 5). Porewater DOC varies disproportionally with OC. The distinct feature of vertical DOC profile is the non-uniform distribution of concentration with mangrove age in contrast with the relatively uniform profile at the bare sediments. Porewater DOC in non-vegetated sediments is known to be primarily controlled by oxygen availability and the presence of microphytobenthos that could drive porewater dynamics via OC leaching compared to mangrove sediments. At the vegetated sites, fluorescence and hydrophobic DOC with high molecular weights are known to drastically increase in anoxic coastal porewaters (Komada et al., 2004; Marchand et al., 2006). At the vegetated sites, porewater DOC showed higher concentrations with depth that is most likely a reflection of the net effect of diagenesis, subsurface transport, and partial control via root uptake and release. Low surface DOC might be the reason of higher mixing and dilution with overlying water with low DOC concentration (Table S2). Though the salinity profile is not exactly like DOC, the salinity peaks at the sub-surface depth, thereby creating a uniform pattern followed at AM and MM. This may lead to the hypothesis that water absorption by roots at the upper sediment may be augmented by the presence of radial mangrove roots leading to an increase of salinity at some sites and gravitational percolation of salt water and DOC to greater depths. However, unlike porewater salinity, which is lower in the wet season than dry season due to rainwater dilution, DOC did not vary seasonally in the adult mangrove and mature mangrove sites suggesting perennial source and retention in the sediment. Porewater profiles of salinity and DOC in mangrove sediments are very rare in literature. One such comprehensive dataset by Marchand et al. (2006) in the French Guiana reported similar findings of higher

DOC at greater depth with mangrove aging but no direct correlation with other co-variables. Like shown many years ago by Marchand et al. (2006), the influence of mangrove productivity and root activity on the DOC vertical profile may be evident also for this study, however, detailed research is necessary to understand such relationships.

### 4.4 Increase of organic carbon with mangrove development

The average TOC concentration in the top 50 cm mangrove sediments (PM to MM) in Bakhawan Ecopark is lower ($2.5\pm1.8$ µmol mg$^{-1}$) than the Indo-Pacific regions ($9.9\pm5.2$ µmol mg$^{-1}$; Donato et al., 2011), but comparable with the global mean (1.7 µmol mg$^{-1}$; Kristensen et al., 2008) and other restored mangroves in SE Asia (e.g., $2.2\pm0.05$ µmol mg$^{-1}$ in Vietnam; Dung et al., 2016) and China ($4.2\pm0.3$ µmol mg$^{-1}$; Nam et al., 2016). The absence of peat organic layer at the sampled sites and fast decomposition observed in separate $CO_2$ emission measurements (benthic emission: 8 mmol m$^{-2}$ hr$^{-1}$ nearly at the upper limit of global range 0.25 to 10.4 mmol m$^{-2}$ h$^{-1}$, Bouillon et al., 2008) are the possible reasons behind such low to moderate OC in the sediments. Besides this, the rapid flushing out of POC favored by the low-lying gentle elevation could account for the OC-poor state of the system. Isotope evidence of POC further indicates that it was sourced from the eroded mangrove soil composed of litter debris (mean around –25.0‰ for salinity 0–25, Table S1) coming from the upstream and mangrove sites that flush water away via the channel to the offshore (–25.0 ‰ at salinity 33) during ebb tide.

The present result shows that the development of restored mangrove forests could improve sediment OC as indicated by a clear progression in sediment TOC among the planted mangroves of different developmental stages leading toward soil maturity. The accumulation of OC in the mangrove sediments may be attributed to the increase in belowground root expansion with stand age (Salmo et al., 2013). Because of restricted water exchange with seawater, mature *Rhizophora* stands accumulate higher mangrove-derived material (leaf litter and decomposing fine roots) than the young and pioneering stands, therefore showing higher TOC. For the adult mangrove stands (AM), despite their moderate biomass, oxic conditions (positive ORP) may have favored OM decomposition in the wet season resulting in lower TOC than at MM. Such non-linear correlation of OC with mangrove chronosequence is in line with other works in the restored (Lunstrum and Chen, 2014; Van Hieu et al., 2017) and intact forests (Lovelock et al., 2010; Marchand et al., 2017).

Like TOC concentration, OC stocks in sediments gradually increased with chronosequence, which is consistent with other recent studies on mangrove plantations (Lunstrum and Chen, 2014; Van Hieu et al., 2017; Wang et al., 2021). Salmo et al., (2013) previously reported higher above-ground biomass (AGB) in 17-yr old *Rhizophora* than the 12-yr old stands (101.8 versus 51.4 Mg ha$^{-1}$) at the Bakhawan Ecopark. Fine root production has also been found to increase with AGB and contribute to sediment OC stocks more in mature mangroves (Zhang et al., 2021). On a global basis, OC stocks are mostly reported for sediment cores of 1-m depth. If the core depth is normalized to 50 cm, our results still give comparable estimates (3–77 Mg C ha$^{-1}$) to the limited assessments for the restored mangrove park (Table 3). Sediment OC stock is largely dependent on vegetation biomass and the extent of litter input. Salmo et al. (2013) reported that 17- and 18-year-old *Rhizophora* stands at the Bakhawan Ecopark had 30–40 % lower AGB compared to natural mangroves (150 t ha$^{-1}$) while 50-year-old stands had similar AGB (132 t ha$^{-1}$) with natural mangroves. At the Bakhawan Ecopark, OC stocks have been shown to vary with distance to the seaward and landward edges (Kauffman

et al., 2011; Wang et al., 2013; Chatting et al., 2020) due to tidally-driven nutrient cycling, OM retention and transport of allochthonous material. River-dominated mangrove settings are known to transport high allochthonous input and deposit mineral sediments that dilute OM and lower OC stocks than marine settings where mangrove-derived OM increase carbon stocks in the sediments (Jennerjahn, 2020). Another reason could be the sediment reworking during restoration or plantation work that can mobilize OC at least from the top 10 cm of the sediment. The rapid turnover may lead to a reduction in OC stocks. From a more general perspective, sediment OC stocks depend on the distribution of fine roots at the adult sites (Noguchi et al., 2020), and roots generally have lower decomposition rates than leaves, favoring C storage partly because of the composition that is relatively rich in recalcitrant material such as suberin and lignin root-derived C (Rasse et al., 2005).

The increase in porewater DOC concentrations observed from the tidal flat to pioneer mangroves, and then to the older mangroves seems to reflect the increasing pattern of bulk OC (Fig. 2j,l). With greater biomass, sediment TOC and porewater DOC concentration successively increased, the latter due to greater leaching of the sizable SOM with age. A similar trend for bulk and dissolved OC was observed in the naturally growing *Avicennia*-dominated mangroves in the Amazonian coastline (Marchand et al., 2006). With the progression of chronosequence, we may infer that the higher the biomass, the higher the TOC, and the higher the DOC concentrations in the shallow sediments.

Organic carbon preservation in marine sediments can be influenced by physical factors such as the association of OC with the surface of sediment mineral particles, also known as OC loading (Keil and Mayer, 2014). Sediments with high SSA tend to be rich in clay, iron and aluminum and store greater amounts of OM than low-SSA soils due to intimate organo-mineral association (Mayer,1994). Therefore, OC loading can be considered as a proxy for blue carbon preservation and supply in the marine system, although there has been no account of such result for mangroves worldwide. Our estimates of OC loading for mangrove sediments and tidal flats (mean: 152 and 25 $\mu$mol C m$^{-2}$) are within the range of marine and costal observations (Table 3), but much lower than in high altitude soils where mineral phases are fully covered by rich-OM with high OC% (300–780 $\mu$mol C m$^{-2}$, 5 to 10%; Wagai et al., 2009). At the Bakhawan Ecopark, the variability of OC loading between mangrove sediments and tidal flats can be explained by the difference in the spatial extent of the individual sampled sites. For example, when SSA is greater than 15 m$^2$ g$^{-1}$ (at AM and MM), a significant negative correlation was observed between OC and SSA (r$^2$ = 0.95, p <0.01, n = 7, figure not shown) indicating that the net accumulation rate of OC in these sites are not dependent on mineral particles but on the mangrove-derived supply of OM. Whereas at BS with mean SSA <15 m$^2$ g$^{-1}$, a quasi-significant positive relationship between these two variables suggests the possible role of the physical sorption of OC in the riverine sediment mineral matrices for stabilization and sequestration of organic carbon (r$^2$ = 0.50, n = 6). Therefore, the sediments in the mangroves do not share a common OC sequestration mechanism as with the pelagic continental shelf and seagrass sediments.

The burial of OC in the sediment strongly depends on many environmental conditions such as mangrove forest productivity, deforestation and degradation rate, sediment accretion, topography, tidal regime, and bioturbation activities (Alongi, 2014; Pérez et al., 2018). Significant exponential and/or linear increase of OC stocks with early mangrove age (until YM) could be the result of longer time duration since the early colonization of plants (Fig. 7a).

However, the correlation between the standing stocks of sediment OC with aging at the adult/mature stands are essentially not significant, indicating that the size of the sediment OC pool at these two sites might be constrained by some biological or geophysical factors. The rates of OC accumulation defined by their respective slopes in the exponential curve (BS to MM) and linear curve (YM to MM) at the individual sites of 50-cm depth (5.9–33 mol m$^{-2}$ yr$^{-1}$, and 15.93 mol m$^{-2}$ yr$^{-1}$, respectively, Figs. 7a,b) are well within the ranges by chronosequential analysis of 1-m cores in restored (12.7 mol m$^{-2}$ yr$^{-1}$, Lunstrum and Chen, 2014), conserved (14 mol m$^{-2}$ yr$^{-1}$, Pe´rez, et al., 2018), intact (6 to 40 mol m$^{-2}$ yr$^{-1}$, Marchand et al., 2017), or encroaching mangroves (19 mol m$^{-2}$ yr$^{-1}$, Kelleway et al., 2016). System-specific variabilities such as sedimentation rate, decomposition rate, rate of litterfall, and root production may cause such differences in CAR among the reported mangroves. It is noted that OC% reported for the mangrove soils were on average 2 times higher than the Bakhawan sites due to high terrestrial inputs to this forest driven by historical land use changes.

Finally, it is important to highlight that the progression patterns of C stocks and/or CAR with mangrove age are observed out of total of seven cores only, and the present dataset doesn't have enough numbers to test the effects of many other variables to the relationship between C stocks, CAR, and mangrove age. Such a relationship could be changed by environmental factors such as topography, hydrodynamics, geomorphology, and biodiversity. We also found a significant effect of soil depth on OC concentration (Table 1). However, as mentioned in section 4.1, environmental variabilities like hydrological processes do not vary largely among the sites and have been relatively stable over ~30 years after restoration, and biomass development follows a similar trajectory of soil salinity, plantation spacing, and species richness. Therefore, the results or conclusion of this study might not significantly change due to the lack of replicates of sediment cores from the restored sites. Nonetheless, acknowledging this as a limitation of the study, we further recommend that several cores are required for drawing robust carbon and age relationships, especially for regions where environmental variabilities can be significant drivers of these relationships.

**4.5 Implication of blue carbon chronosequence**

There is a crucial need to improve the scientific understanding of blue C dynamics and to develop an appropriate framework for blue C assessment and monitoring mechanisms for future policy development. Achieving an improved scientific understanding of C sources and stocks and monitoring the changes in accumulation rate at the early development stage and adult stages of mangrove stands would require practical tools and guidance to enable the conduct of proper C analyses. In this study, the supply of OM from mangrove vegetation, benthic algae, and upstream sediment transport are recognized as controls of blue C at the mangrove sites and tidal flats. The impression of more mangrove-derived C input is evident with mangrove development. Such apportionment of OC sources at different mangrove stand ages should be useful to improve our future knowledge on the origin of blue C in REDD+ accountings that is yet to register sedimentary OC within the reduced C emission scheme despite the sediment being recognized as the largest C pool in the mangrove ecosystem (Duarte et al., 2013). Prior knowledge of the sources and characteristics of OC (generally refractory or mangrove-derived and labile or algal-derived) would be beneficial for fostering mangrove plantation programs. Greater mangrove-derived OC accumulation with aging at the Bakhawan Ecopark might suggest long-term storage of the refractory fraction, hence an effective return to the REDD+ strategy.

Any attempt to quantify OC stocks and accumulation rates following a plantation program should be well recorded in relevant carbon accounting programs. For example, the Verified Carbon Standard Methodology ([www.verra.org](http://www.verra.org)) existing for mangrove restoration projects that are certified under the Cleaner Development Mechanism (CDM) program of the United Nations Framework Convention on Climate Change (UNFCCC, 2021) assumes CAR for 0–20 years mangrove as 4 mol m$^{-2}$ yr$^{-1}$ after plantation (Lunstrum and Chen, 2014), which is 4 times lower than our chronosequence based estimate (~16 mol m$^{-2}$ yr$^{-1}$, linear slope for YM to MM stands, figure not shown). For the Bakhawan Ecopark with mangrove coverage of 121 ha, the total organic carbon stocks and accumulation rates in the vegetated soil are estimated to be 2795–11500 Mg C ha$^{-1}$, and 72–304 Mg C yr$^{-1}$, respectively. Carbon stocks in the above-ground biomass were 2744–5499 Mg C ha$^{-1}$ for the younger and adult sites (derived from literature data by Salmo et al., 2013, using biomass to C conversion factor of 45% from IPCC, 2013). However, if the IPCC default values are used to compute the total C stocks in the sediment and above-ground biomass (386, and 92–192 Mg C ha$^{-1}$, respectively), significant differences can be observed between the observed and computed results (5009–10454, and 46700 Mg C ha$^{-1}$, respectively). Considering that the IPCC default values were derived from intact forest while our observed data were taken from restored sites, direct comparisons as such have potential biases. Similar discrepancies are also present for emission fluxes. In IPCC Tier 1, the default $CO_2$ emission factor from the tidal flat was set to be 0, while for the planted/rehabilitated mangroves was –14 mol m$^{-2}$ yr$^{-1}$ (negative value meaning accumulation). The latter is in line with the older mangroves at Bakhawan Ecopark, but not with each developmental stage. Such an assumption may lead to severely underestimated actual blue carbon sink capacities of the mangroves and consequently, carbon emission values. Therefore, a differential yet steady trend of blue C potential based on short-term chronosequence can help define the utility of mangrove restoration efforts.

## 5. Conclusion

This study is a first attempt to apply chronosequence (or space-for-time substitution) approach to evaluate the distribution and accumulation rate of carbon in a 30-year-old (maximum age) restored mangrove forest. From this study, it is clearly seen that mangrove tree development coincided with sediment OC concentration and accumulation, hence mangrove plantations are expected to accelerate OC sequestration at the early plantation stage. Source apportionment of sedimentary OM suggests higher contribution of mangrove vegetation at higher elevations and riverine POM as well as benthic algae down in the tidal flat. The accumulation of OC in the sediment may be attributed to the increases in belowground root expansion with stand age. These chronosequence-based estimates of OC stocks and accumulation rates can be useful references for setting up carbon accounting in *Rhizophora*-dominated mangrove restoration projects.

**Acknowledgement**

We are grateful to the Japan International Cooperation Agency (JICA) and Japan Science and Technology Agency (JST) through the Science and Technology Research Partnership for Sustainable Development Program (SATREPS) for financially supporting the Project "Comprehensive Assessment and Conservation of Blue Carbon Ecosystems and

their Services in the Coral Triangle (Blue*CARES*)". We thank Jesus Abad, John Michael Aguilar, Dominic Bautista, Bryan C. Hernandez, and Mr. Tsuyoshi Kanda for their assistance during field surveys. We are grateful for the overall support given by the University of the Philippines, Diliman, and Aklan State University to the project. We are thankful to our BlueCARES colleague Dr. Charissa Ferrera for the support in language edits. Finally, authors sincerely thank
AE (Jack Middelburg) and the reviewers for their constructive comments that have greatly improved the revised version of the manuscript.

**Competing interests**

The authors declared no potential conflict of interest.

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

**Figure legends**

**Figure 1: Map of the study area. BS (Bare sediment); PM (Pioneer mangrove, 3-month); YM (Young mangrove, 10-yr); AM (Adult mangrove, 20-yr); MM (Mature mangrove, 30-yr). The circle shows presence of long-tailed sandbar between AM and YM site. The north-west inland part and bank of the Aklan River are the dominant places for the naturally occurring mangroves.**

**Figure 2: Box plot diagrams of physical and biogeochemical parameters in the sediment subsamples according to mangrove development.**

**Figure 3: Vertical profiles of sedimentary carbon properties during wet and dry season.**

**Figure 4: Vertical distribution of sediment surface area and OC loading during wet and dry season**

**Figure 5: Vertical profiles of porewater DOC during wet and dry season**

**Figure 6: Source identification of sedimentary organic matter using endmember carbon stable isotope ratio and OC:TN.**

**Figure 7: a) The relationship between mangrove age (*Age*) and carbon stock, where the curve was drawn based on an exponential function model in Eq. 2: *OC* = 171.07exp(0.03558\**Age*), $R^2$ = 0.9873. The gray band means 95% confidence interval. (b) The relatioinship between mangrove age and carbon accumulation rate (CAR) on the basis of the exponential model (see Eq. 3).**

**Table 1: The significance of effects of type and depth to bulk TOC, porewater DOC, bulk OC/TN ratio $\delta^{13}$C and $\delta^{15}$N using ANOVA on the basis of the generalized additive model (GAM). In the parameter of depth, the approximate significance of smooth term is shown.**

| Effect | TOC | DOC | OC:TN | $\delta^{13}$C | $\delta^{15}$N |
|--------|---------|------|---------|---------|---------|
| **Type** | < 0.001 | 0.002 | < 0.001 | < 0.001 | 0.005 |
| **Depth** | 0.0002 | 0.13 | 0.98 | 0.40 | 0.73 |

**Table 2. Contribution (%) of endmember sources to sediment organic matter. POM: Particulate Organic Matter, MPB: Microphytobenthos AM: adult mangrove, MM: mature mangrove, PM: pioneer mangrove, YM: young mangrove. Bare sediment or BS was set as reference level. Mean±SD (mean at 95% confidence level)**

| Sampling site | Mangrove leaf (%) | Marine POM (%) | MPB (%) | River POM (%) |
|---------------|-------------------|----------------|---------|---------------|
| BS | 18±14 (45) | 18±21 (67) | 6±9 (25) | 58±28 (92) |
| PM | 32±17 (63) | 18±19 (59) | 7±9 (29) | 43±24 (83) |
| YM | 53±16 (77) | 13±11 (40) | 5±6 (18) | 29±18 (60) |
| AM | 41±14 (64) | 14±12 (40) | 5±4 (16) | 39±18 (68) |
| MM | 58±19 (86) | 11±9 (31) | 5±4 (13) | 29±19 (66) |

**Table 3. Comparative results on carbon stocks (Mg C ha$^{-1}$) in restored mangroves of known ages, and organic carbon loading (µmol C m$^{-2}$) in mangroves with other marine settings.**

| Location | Age (year) | Dominant species | Soil C stock Mg C ha$^{-1}$ | OC loading µmol C m$^{-2}$ | Reference |
|---|---|---|---|---|---|
| Philippines, Panay | | | | | |
| Bakhawan EP | 0 | No vegetation | 3.1-24.3 | 4-58 | This study |
| Bare sediment Pioneer | 0.25 | *Avicennia marina, Rhizophora spp* | 21.4 | - | This study |
| Young | 10 | *Rhizophora apiculata* | 23.1 | - | This study |
| Adult | 20 | *Rhizophora apiculata* | 36.9-46.4 | 42-148 | This study |
| Mature | 30 | *Rhizophora apiculata* | 61.3-93.5 | 68-380 | This study |
| Planted | - | *Rhizophora sp.* | - | 310-1140 | Unpublished data, Miyajima et al. |
| Naturally recovered | - | *Avicennia Rumphiana* | - | 57-640 | Unpublished data, Miyajima et al. |
| North-central Vietnam | 0 - 27 | *Kandelia obovata* | 54 - 84 | - | Van Hieu et al., 2017 |
| Pichavaram, India | 12-21 | *Rhizophora spp* | 41-94 | - | Gnanamoorthy et al., 2019 |
| Bhitarkanika, India | 5 | *Kandelia candel* | 38 | - | Bhomia et al., 2016 |
| Sulawesi, Indonesia | >10 | *Rhizophora apiculata* | 150-300 | - | Cameron et al., 2019 |
| Continental margin | - | - | - | 40-80 | Mayer, 1994 |
| Vegetated marine sediment | - | - | - | 56-67 | Miyajima and Hamaguchi, 2017 |
| Floodplain | - | - | - | 16-42 | Goni et al., 2014 |



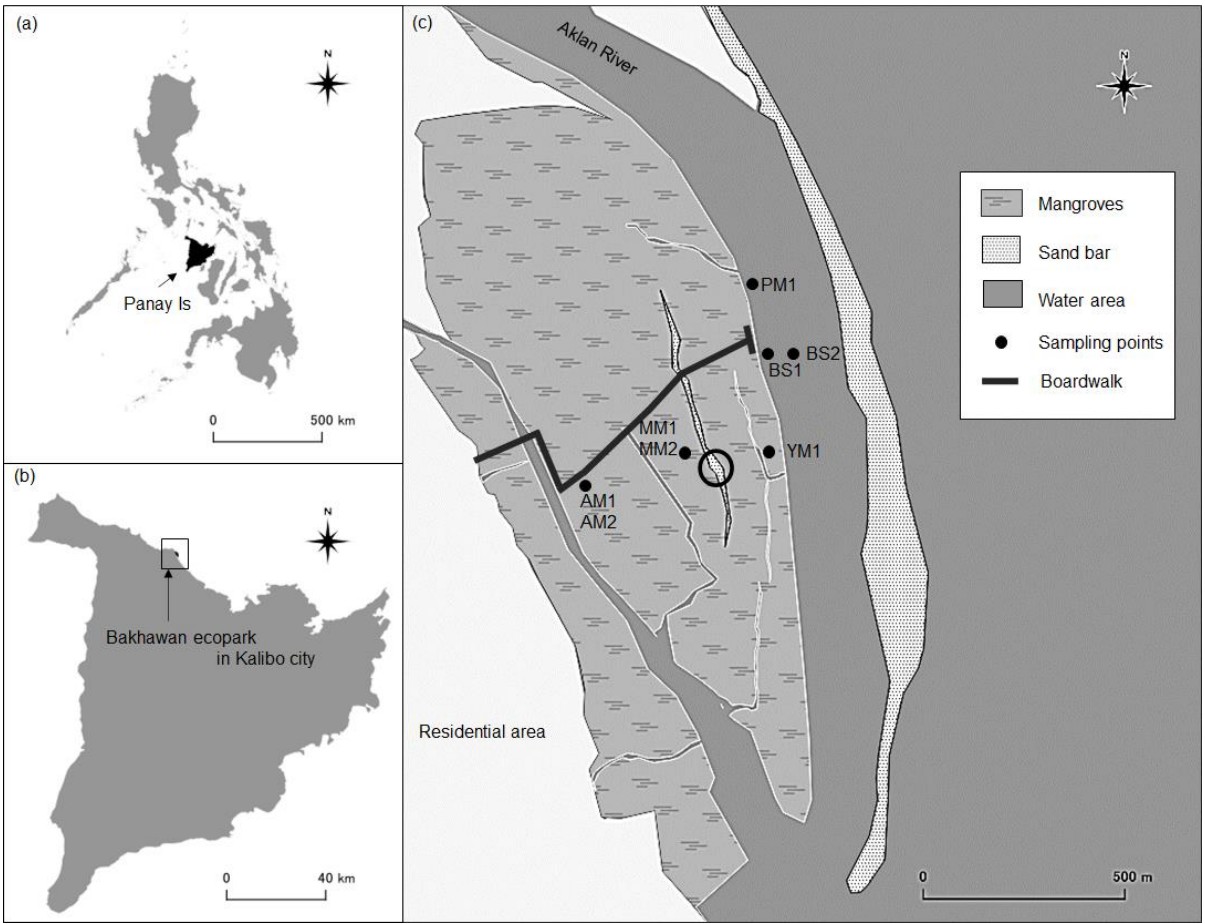

**Fig. 1.**



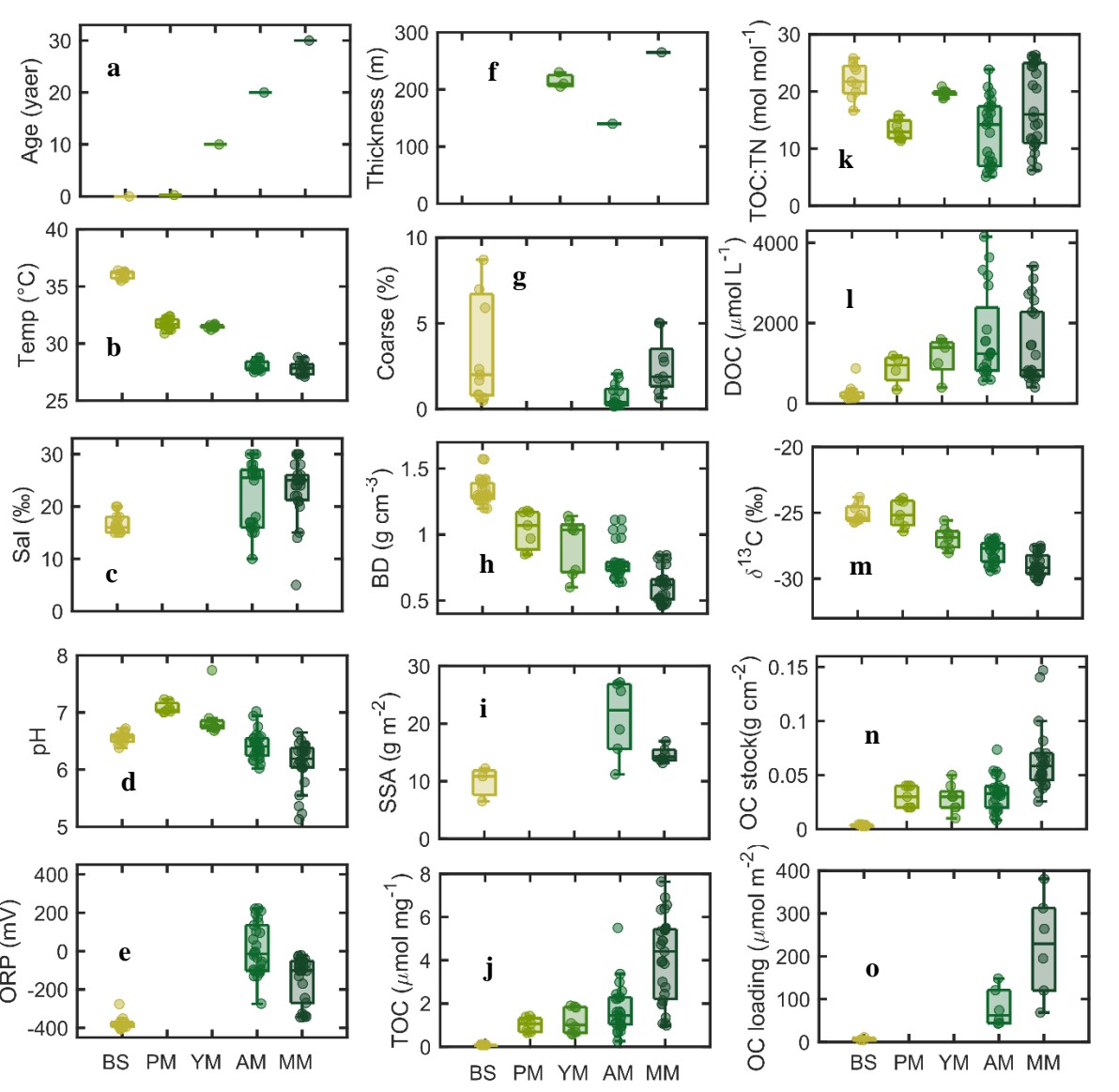

**Fig. 2.**


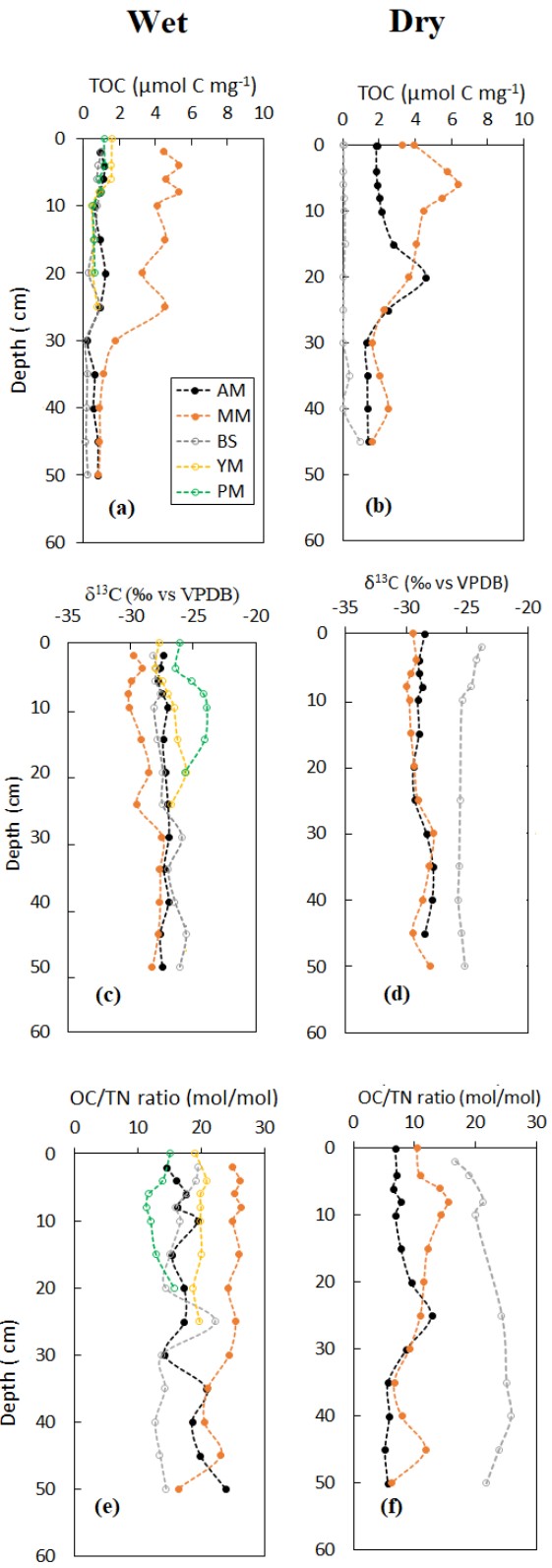

**Fig. 3.**

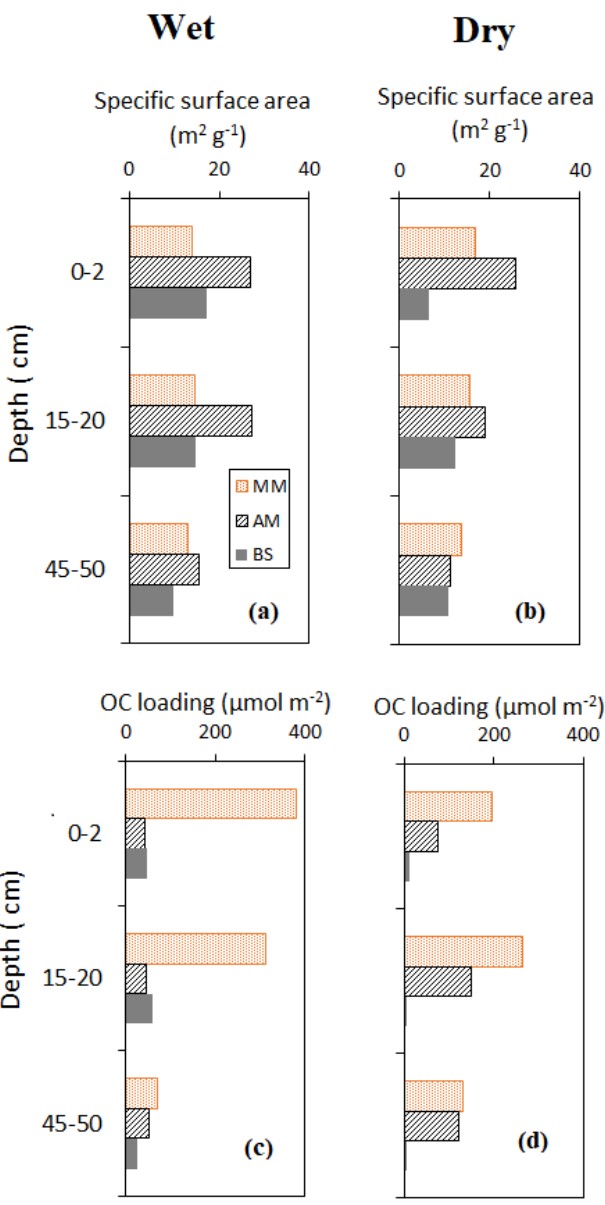


**Fig. 4.**



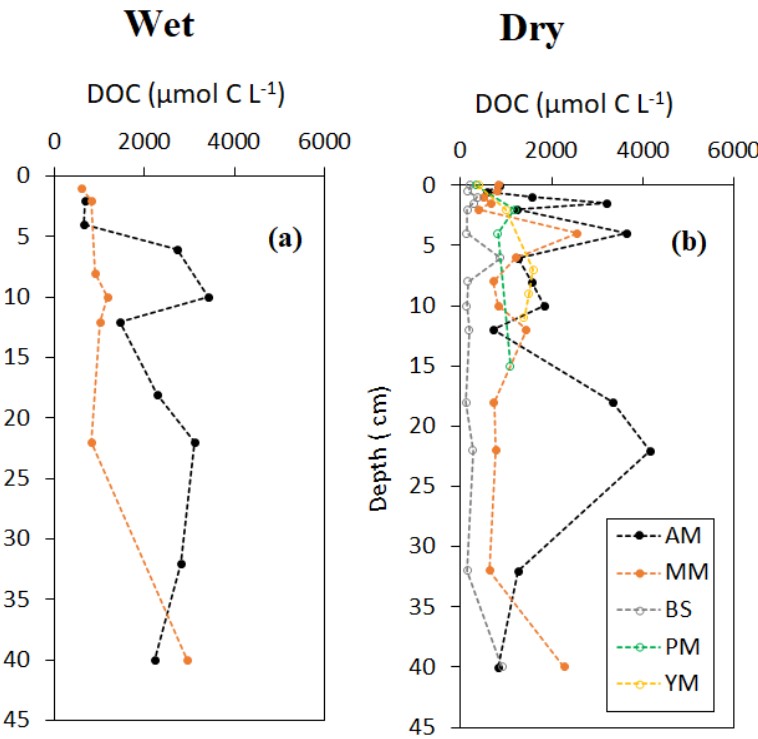

**Fig. 5.**




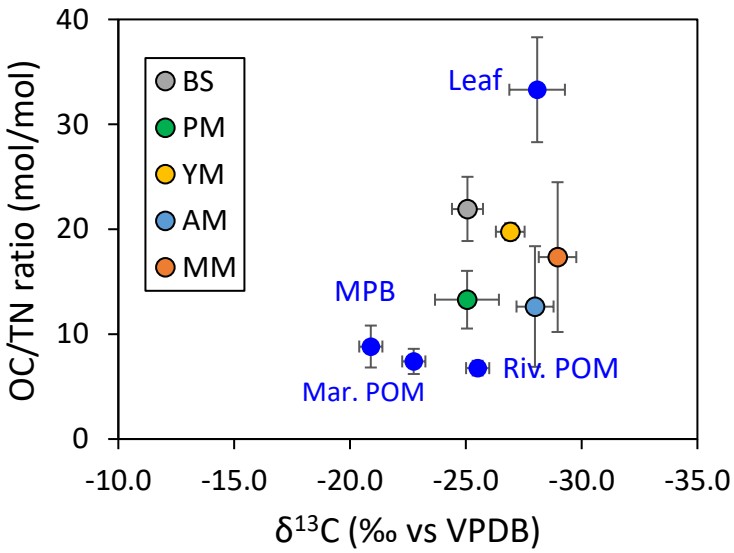

**Fig. 6.**





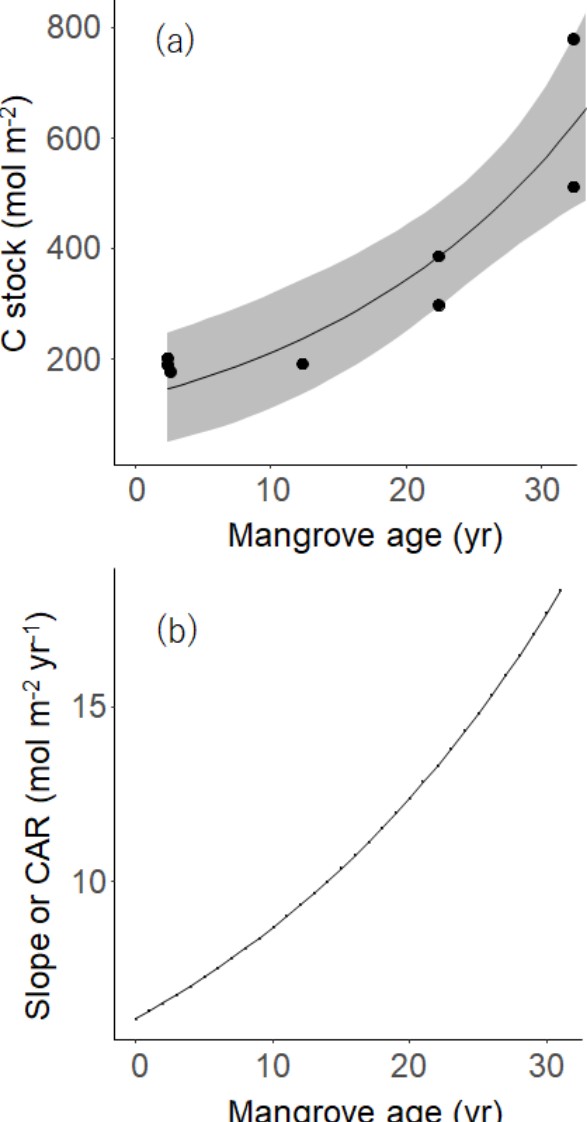

**Fig. 7.**
