# Peer review of "Sedimentary blue carbon dynamics based on chronosequential observations in a tropical restored mangrove forest"

_Biogeosciences, 2021_

## Author Response (AR1)

Response to comments

**Reviewer#1**

This is an interesting and relevant study that applies a chrono-sequence approach to study carbon accumulation in relation to time since mangrove restoration. The study reports that, based on isotopic signatures, the contribution of mangrove plant material was higher at older sites while younger sites have a higher contribution from riverine inputs. In general the paper is nicely written and uses standard physico-chemical analysis.

My concern is that the sample size is very limited; only 5 cores were used to study the chrono-sequence and no replicate cores were taken. I would agree that this can show some trends and differences between the ages, but a robust statistical quantification or test of the hypothesis is challenging. The study does not report results of statistical tests or uncertainty ranges. In short, I found it difficult to understand heterogeneity and uncertainty and this is really important as it defines the limits of interpretations. In my opinion, the authors should address this basic but critical issue.

Reply: Thank you for valuable suggestion regarding statistical validation with uncertainty range. In the revised ms, the 95% confidence interval was added for the Age – C stock relationship (Fig. 8). The results were slightly changed since the previous regression based on exponential function was done with a log-transformation of independent variable, which was not suitable for the comparison with a linear function model. In the present study, the exponential function model was fitted using nonlinear least square method with nls function of R and 95% confidence interval was added using predFit function of investr package of R. Conclusion was not changed but the uncertainty range of the model analysis has been clearly shown in the current ms. Furthermore, MixSIAR model enables to produce uncertainty ranges in contribution % of different end member to OC sources which is another very vital aspect of this research.

[Figure]

Figure 8: a) The relationship between mangrove age (*Age*) and carbon stock (*OC*), where the curve was drawn based on an exponential function model in Eq. 2: $OC = 171.07\exp(0.03558*Age)$, $R^2 = 0.9873$. The gray band means 95% confidence interval. (b) The relatioinship between mangrove age and carbon accumulation rate (CAR) on the basis of the exponential model (see Eq. 3).

Line 78: could use an additional sentence that links problem statement with hypothesis

Reply: A new sentences on problem statement is added

In this study, we address the question about how chrono-sequential observation in a restored mangrove forest could guide us achieving improved scientific understanding on C sources, stocks and to monitor the changes in accumulation rate at the early development stage and adult stages.

Line 127: This seasonal collection doesn't match with what is shown in Figure 1 and Figure 8(ie BS but not YM?) Please check

Reply: Thank you for the comment. The text is corrected as

A total of 8 cores were retrieved during the survey period with seasonal collection made at BS, AM, and MM site (dry and wet, total cores 6) and wet season collection from PM and YM site (one each).

Line 227: Are this mean values for the whole core? If so, add this to the figure caption.

Reply: Figure 2 caption has been edited by mentioning (mean±SD)

Line 400+: Consider adding an overview table where you summarize literature and your own data

Reply: Thank you for the recommendation. As proposed, a new table on comparative stock (Mg C ha$^{-1}$) and organic carbon loading (µmol C m$^{-2}$) has been provided.

Table 2. Comparative results on carbon stock (Mg C ha$^{-1}$) in restored mangroves of known ages, and organic carbon loading (µmol C m$^{-2}$) in mangrove with other marine settings.

| Location | Age (year) | Dominant species | Soil C stock Mg C ha$^{-1}$ | OC loading µmol C m$^{-2}$ | Reference |
|---|---|---|---|---|---|
| Philippines, Panay | | | | | |
| Bakhanwan EP | 0 | No vegetation | 3.1-24.3 | 4-58 | This study |
| Bare sediment | | | | | |
| Pioneer | 0.25 | *Avicennia marina, Rhizophora spp* | 21.4 | - | This study |
| Young | 10 | *Rhizophora apiculata* | 23.1 | - | This study |
| Adult | 20 | *Rhizophora apiculata* | 36.9-46.4 | 42-148 | This study |
| Mature | 30 | *Rhizophora apiculata* | 61.3-93.5 | 68-380 | This study |
| Planted | - | *Rhizophora sp.* | - | 310-1140 | Unpublished data, Miyajima et al. |
| Naturally recovered | - | *Avicennia Rumphiana* | - | 57-640 | Unpublished data, Miyajima et al. |
| North-central Vietnam | 0 - 27 | *Kandelia obovata* | 54 - 84 | - | Van Hieu et al., 2017 |
| Pichavaram, India | 12-21 | *Rhizophora spp* | 41-94 | - | Gnanamoorthy et al., 2019 |
| Bhitarkanika, India | 5 | *Kandelia candel* | 38 | - | Bhomia et al., 2016 |
| Sulawesi, Indonesia | >10 | *Rhizophora apiculata* | 150-300 | - | Cameron et al., 2019 |
| Continental margin | - | - | - | 40-80 | Mayer, 1994 |
| Vegetated marine sediment | - | - | - | 56-67 | Miyajima and Hamaguchi, 2017 |
| Floodplain | - | - | - | 16-42 | Goni et al., 2014 |

Line 450: I like this section as it justifies the chronosequence approach. This could be presented a bit earlier in the ms?

Reply: We agreed on that recommendation, and accordingly moved section 4.4.1 as 4.1 and stat the discussion by giving justification of chrono sequence approach.

**Reviewer#2**

Overarching comments: The study investigates the carbon dynamics in mangrove stands of differing ages. This study provides useful information on the differences in carbon stocks and potential source of input throughout a mangrove stand's maturation. In general the study is well thought out and presented. Authors frame the study well and provide sound reasoning for various data collected. My main concern is the number of soil cores taken from different mangrove stand ages. A total of eight samples are used to compare between 4 different age classes and between 2 seasons. This is only one sample per unique experimental condition. That is not enough to perform the rigorous statistical comparison this study deserves. Also, the English language has many minor grammatical errors and should be edited by a native speaker.

Reply: Thank you for the comment. We agree that sample sizes are small, and we did not take replication of soil samples at each site, and can't evaluate uncertainty for carbon stocks at each individual condition, however, we calculated the uncertainty of the predicted CAR which is one of the main items to be evaluated in the present study.

The results were slightly changed since the previous regression based on exponential function was done with a log-transformation of independent variable, which was not suitable for the comparison with a linear function model. In the present study, the exponential function model was fitted using nonlinear least square method with nls function of R and 95% confidence interval was added using predFit function of investr package of R. Conclusion was not changed but the uncertainty range of the model analysis has been clearly shown in the current ms.

Furthermore, MixSIAR model enables to produce uncertainty ranges in contribution % of different end member to OC sources which is another very vital aspect of this research.

We are thankful to our BlueCARES colleague and ocean science expert Dr. Charissa Ferrera for supporting in language edits throughout the manuscript.

Line 21: Drop the word "the". It should read "estuary in Panay Island, Philippines" Done

Line 21: I have never come across the term "endmember source apportionment." What does it mean?

Reply: Its rewritten as "…source apportionment of multiple endmembers in the"

Source apportionment is a practice to identify different endmembers as a tracer. For instance, a coastal sedimentary OC pool can be mixture sources like phytoplankton, C3 plant, C4 plant and so on, each of which are endmember and contributor to the total pool.

Lines 27 to 29: This sentence does not convey the importance of the study or the wider implications of the findings well.

Change to "Hence, sediment of relatively young mangrove forest appears to be significant C sink, and short term chrono-sequence based method can efficiently define the importance of mangrove restoration program as potential carbon sequestration pathway."

Line 32: Drop the word "the". It should read "Mangroves, located around tropical and…" Done

Line 34: Should be plural when referring to "carbon stocks". Please change here and throughout.

Done as suggested throughout the manuscript

Line 35: The proportion of sediment organic carbon found in mangroves can vary much more widely than 73-79%. I suggest the authors use a few more citations for this sentence to clarify this to the readers.

Added two new citations in support of the statement (Alongi, 2020; Hatje et al., 2020; Walcker et al., 2018).

Line 39: Organic matter mangrove soil depths can extend much deeper than 3m. I suggest the authors include that information here with a citation

Reply: New sentence added "Sediment depth in the undisturbed mangrove forest sites can often exceed 3 m (Elwin et al., 2019)."

Line 44: "2 to 8 times lower" doesn't make sense. It should be one eighth or one half.

Reply: Corrected as "…1/8$^{th}$ to 1/2$^{th}$ lower…"

Line 58: SFT has already been defined one line previously, the authors don't need to do it again

Reply: Simplified as

In this study, an evaluation based on a type of 'natural experiment' or chrono-sequence (a.k.a. "space-for-time-substitution" or SFT; Pickett, 1989) is made to a relatively younger site (e.g., Bakhawan Ecopark, Philippines, examined in this study) where to fulfill conditions for chrono-sequence, all environmental and biological conditions of the experimental sites must be identical except for the age, and species diversity is low (Nilsson and Wilson, 1991; Walker et al., 2010).

Line 75: I've never come across the term "end-member" what does it mean?

Reply: As mentioned before, endmembers are used as a tracer for source apportionment in OC pool. For instance, a coastal sedimentary OC pool can be mixture sources like phytoplankton, C3 plant, C4 plant and so on, each of which are endmember and contributor to the total pool.

References are:

https://en.wikipedia.org/wiki/Endmember

https://www.sciencedirect.com/topics/engineering/endmembers

Lines 114-116: Are the authors assuming that all the trees in each section of mangrove (PM, YM, AM and MM) are the same? If so I suggest they are explicit with that assumption.

Reply: With respect to species diversity, its mentioned in the text as:

*Rhizophora apiculata* is the dominant species at YM, AM and MM, while mixed mangroves (*Avicennia* and *Rhizophora* sp.) are found at PM.

With respect to age they are different as also mentioned in the beginning

Sediment sampling locations were categorized according to mangrove ageing as bare sediment (BS, 0-yr), pioneer mangroves (PM, 0.25-yr), young mangrove (YM, 10-yr), adult mangrove (AM, 20-yr) and mature mangroves (MM, 30-yr).

Lines 124 to 134: It is not clear what these cores were taken for. Was it to perform isotopic analysis or to measure soil C stocks? I suggest the authors mention the purpose of these cores like they have on line 135 for porewater sampling cores.

Reply: The purpose of sediment collection is clarified now

Within 3-4 hours after collection, sediment samples were kept in styrofoam box and brought to the laboratory for analyzing of bulk density, specific surface area (SSA), concentrations and isotope ratios of carbon and nitrogen, and carbon stock. Analyses took place normally within a month or maximum two after the collection.

Lines 125 to 127: Is my understanding correct that one core was taken from each forest age type during each season? This is not enough replication to conduct adequate statistical comparison between forest age groups. Especially as the soil thickness is only 50cm, OM and C concentrations are known to vary significantly in shallower soil horizons.

Reply: We agree that sample sizes are small, and we did not take replication of soil samples at each site, and can't evaluate uncertainty for carbon stocks at each individual condition, however, we calculated the uncertainty of the predicted CAR which is one of the main items to be evaluated in the present study.

The results were slightly changed since the previous regression based on exponential function was done with a log-transformation of independent variable, which was not suitable for the comparison with a linear function model. In the present study, the exponential function model was fitted using nonlinear least square method with nls function of R and 95% confidence interval was added using predFit function of investr package of R. Conclusion was not changed but the uncertainty range of the model analysis has been clearly shown in the current ms.

Furthermore, MixSIAR model enables to produce uncertainty ranges in contribution % of different end member to OC sources which is another very vital aspect of this research.

Line 127: what about BS and PM sites?

Reply: Corrected as: A total of 8 cores were retrieved during the survey period with seasonal collection made at BS, AM, and MM site (dry and wet, total cores 6) and wet season collection from PM and YM site (one each).

Line 130: Average depth and soil depth in all sites should be reported in the results section.

Reply: Total sediment depth for each site has been explicitly mentioned now in the M&M so that we don't repeat same value in the result section:

Total sample depth at BS, AM and MM were always 50 cm, while for PM and YM they were 20 cm and 25 cm, respectively.

Line 178: Why did authors only measure soil C stocks in the top 50cm depth?

Reply: Sediment coring was easier till 50cm at BS, AM, and MM, however, after that it was hard to penetrate further, where as at PM and YM, sediment was even harder to reach to 50 cm.

Line 190: Where are a and b derived from? Are they coefficients from calculating a line of best fit? If so, I suggest the authors make that clear.

Reply: Clarified as "…where *a* and *b* are constants determined from the best-fit exponential relationship"

Equations 2 and 3: Do the authors have figures for these curves that were fit? It would be good for this info to be included somewhere in the supplementary info for readers to see.

Reply: In Fig. 8 shows the best-fit curve for both the equations.

Lines 230 to 240: This paragraph is very data dense and mentions a lot of ranges of data between forest age groups, hence is difficult to follow. I suggest the authors create a summary table for the data explained in this paragraph, it will be much clearer and condense the text in the results section.

Reply: Since these data are captured already in Fig. 2, we believe showing them again in tabulated form would be redundant. However, the paragraph is streamlined.

Line 245: Doe this value carry any uncertainty? How many measurements was this based off?

Reply: Edited as

 Minimum pH was recorded at the top 10cm depth of the mature mangroves in the wet season (5.41±0.26)

Lines 284 to 285: I would not call the BD values in this study different to those reported by Donato et al. 2011. To me these are comparable, especially as in some other (low OC) mangroves BD can reach up to 2.00 g cm-3.

Reply: Agreed. Revised as

Bulk densities (BD) at the sampling sites (0.3 to 1.3 g cm$^{-3}$) were comparable with  the reported BD values across mangrove soils of the Indo-Pacific regions (0.20 to 0.92 g cm$^{-3}$, Donato et al., 2011), with sand fractions dominating the lower intertidal zone

 Line 785, Fig. 2: The information in the upper left most panel is a repeat of what was explained in the text. I don't think it is needed. Also it is interesting that pioneer and young mangrove have comparable OC

stocks. How many replicates was this data based on? Does the YM for this panel have any error bars? Was it just one replicate?

Reply: Although mangrove development with age has been mentioned before in the text, we believe showing that trend in Fig 2 which is otherwise very much result oriented would help interpreting data from chrono-sequence perspective. Furthermore, removing that would break compositeness of the fig. 2.

Reply: Single core was collected from each of PM and YM site. Hence stock data does not have uncertainty range.

A total of 8 cores were retrieved during the survey period with seasonal collection made at BS, AM, and MM site (dry and wet, total cores 6) and wet season collection from PM and YM site (one each).

Line 790, Fig. 4: Do these values carry any uncertainty? It should be included in this figure. Also, why are there only 3 mangrove categories in this fig, what about pioneer and young mangrove?

Reply: As the vertical profile shown in Fig. 4 captures each depth-wise seasonal results, uncertainty ranges could not be shown for this figure.

Also, because of laboratory logistical reasons, SSA were measured only for BS, AM and MM.

However, this is now explicitly mentioned:

Specific surface areas were measured only for BS, AM and MM cores.

I suggest the authors have a native English speaker red the manuscript and make edits. There are many minor grammatical errors throughout.

Furthermore, MixSIAR model enables to produce uncertainty ranges in contribution % of different end member to OC sources which is another very vital aspect of this research.

**Autor's own edits**

In addition to the corrections based on reviewer's comments, we have recognized few others corrections to make for better refinement of the manuscript. Below is the details of changes made from author's side:

Delete- "..due to their role in climate change mitigation"

Corrected- "…oxidation reduction potential (ORP, Pt-electrode)", also ORP is consistently used, not Eh

Corrected - Within 3-4 hours after collection, sediment samples were kept in styrofoam box and brought to the laboratory for analyzing bulk density, specific surface area (SSA), concentrations and isotope ratios of carbon and nitrogen, and carbon stock

Corrected – In Fig 2. We have added labels (a, b, c,...) to individual panels in Fig. 2 and identify a specific panel in the text using such labels (e.g. "Fig. 2b"), also, marked by "nd" where no data are available, so that atleast they are not misunderstood as zero

Corrected: Sediment thickness unit in Fig 2 is corrected to cm

Corrected: SSA unit is corrected as m$^2$ g$^{-1}$ (or m$^2$/g) in text and in Fig. 2. (it was already good in Fig. 4)

Added in text: "….mean $\delta^{15}$N became more positive with mangrove age ($\delta^{15}$N = –1.15 at the BS to 1.06 ‰ at AM, figure not shown).

Added: Fig S2 caption- Solid and dashed lines represent Mean and SD, respectively.

Correction- We recognize that Fig. 7 is not very straightforward than table S3 while both have same message.  Hence we decided to show Table S3 in the main text and move Fig. 7 as  Supplementary Information. In the revised ms, table S3 becomes Table 2 and Fig 7 becomes Fig S3. The caption of Fig. S3 is revised as

Source apportionment of sedimentary organic matter at different mangrove stages by applying bayesian mixing model with $\delta^{13}$C and OC:TN. This is a density plot against proportion of sources. Here, the total area in each curve exceeds 1.0 since this is Scaled-Density adjusted for a maximum peak of 1.0. MixSIAR outputs the Scaled-Density instead of original density. The Scaled-Density shows a same visual pattern as original density plot.

Deleted: "deposition of pollen"  as we know Pollen does not affect SSA, because most of organic matter is removed in advance of measurement. Revised the sentence as : The latter might have been a result of sediment-stabilizing function of mangroves or influenced by weathering process that transforms sand and silt fractions into clay fractions

Changed: " A sizeable contribution of MPB…" to "A sizeable fraction of OC at the tidal flat was contributed by the marine and benthic algae"

New text: under section 4.3 **Vertical profile of organic carbon**;

" The variation in OC content from the bottom to the top of sediment cores also reflected the change in the proportion of mangrove-derived materials and allochthonous (e.g., benthic algae) organic carbon sources in the sediments and the mineralization of organic matter in mangrove sediment (Tue at al., 2011). "

"This has been shown in other studies that root-derived carbon tends to be accumulated more efficiently as aggregate in SOM pool and contribute largely as potential C source (Xia et al., 2005)."

"Low surface DOC might be the reason of higher mixing and dilution with overlying water of low DOC concentration (Table S1)."

New Text: under 4.4**. Increase of organic carbon with mangrove development**

"and roots generally have lower decomposition rates than leaves favoring C storage partly because of the composition relatively rich in recalcitrant material such as suberin and lignin root-derived C (Rasse et al., 2005)."

Revision: Organic carbon burial part under the section "Accumulation of organic carbon" is merged with the section "Increase of organic carbon with mangrove development", while "Relevance of chrono-sequence approach" has been moved up in the discussion as separate section 4.1 (as suggested by Reviewer).

Next text: For the section **Implication of blue carbon chrono-sequence,** we recognized that we only discussed blue carbon in soils, but we have not mentioned this fact explicitly, e.g. for the case of Bakhawan

Ecopark, what would be the estimate total OC stock and accumulation rate including both soil and biomass OC using literature review? Then may be it is worthwhile to mention them and compare them with, e.g., IPCC default values. Based on this argument, we revised the section by adding below text

"For Bakhawan Ecopark, mangrove coverage of 121 ha, total organic carbon stock and accumulation rate in the vegetated soil was estimated to be 2795 to 11500 Mg C ha$^{-1}$, and 72 to 304 Mg C yr$^{-1}$, respectively. Carbon stock in above ground biomass was 2744 to 5499 Mg C ha$^{-1}$ for the younger and adult sites (derived from the literature data by Salmo et al., 2014, and biomass to C conversion factor of 45%, IPCC 2013). However, if IPCC default values are plugged-in to compute total C stock in soil and above ground biomass (386, and 92-192 Mg C ha-1, respectively), significant differences between observed and computed results are observed (computed: 46700, and 5009 to 10454 Mg C ha$^{-1}$, respectively). Given IPCC default values are derived from the intact forest while our observed data were taken from restored sites, such direct comparison has potential biases."

**Newly added citations**

Elwin A., Bukoski, J.J., Jintana, V., Robinson, E.J.Z., Clark, J.M.: Preservation and recovery of mangrove ecosystem carbon stocks in abandoned shrimp ponds. Sci Rep. 4, 9:18275. doi: 10.1038/s41598-019-54893-6, 2019

Hatje, V., Masqué, P., Patire, V.F., Dórea, A., and Barros, F.: Blue carbon stocks, accumulation rates, and associated spatial variability in Brazilian mangroves, Limnol. Oceanogr., 9999: 1–14, doi: 10.1002/lno.11607, 2020.

Rasse, D.P., Rumpel, C., Dignac, M.-F. Is soil carbon mostly root carbon? Mechanisms for a specific stabilisation. Plant, 269, 341–356, 2005

Tue, N.T., Dung, L.V., Nhuan, M.T., Omori, K.: Carbon storage of a tropical mangrove forest in Mui Ca Mau National Park, Vietnam. Catena, 121: 119–126, 2011

Xia M, Talhelm AF, Pregitzer KS.: Fine roots are the dominant source of recalcitrant plant litter in sugar maple-dominated northern hardwood forests. New Phytol. 208: 715– 726, 2015.

**Deleted citation**

Jiménez-Arias, J.L., Morris, E., Rubio-de-Inglés, M.J., Peralta, G., García-Robledo, E., Corzo, A., and Papaspyrou, S.: Tidal elevation is the key factor modulating burial rates and composition of organic matter in a coastal wetland with multiple habitats, Sci. Total Environ., 724: 138205, doi: 10.1016/j.scitotenv.2020.138205, 2020.

---

## Author Response (AR2)

**Response to comments (second revision)**

Dear Dr. Ray:

Thank you for submitting your revised version to Biogeosciences. Your revised version has been evaluated again by two referees. They agree that your paper addresses an important topic, but they also believe that it needs another round of revision before publication. Specifically, both argue that the limited replication lowers the robustness of your conclusions. They recommended articulating clearer the exploratory nature of your study and the consequences of limited replication for the conclusiveness. The referees also made some additional remarks that need attention (mainly grammar).

On the basis of these reviews, I cannot yet accept this paper for publication and invite you to submit a revised manuscript resolving the last issues remaining.

With best regards,

Jack Middelburg

Response:

Dear Prof. Middelburg,

Thank you very much for the decision. In the revised version, we have addressed all the comments made by the reviewers. Below are some bulleted reflections on the major revision
- ➔ We realized that our dataset is enough to evaluate the effect of age, and this is the most important point of this work. In contrast we believe that the dataset is not sufficient to convincingly evaluate seasonal variations, and seasonal variation is out of the scope of chrono-sequential analysis. Hence, we reanalyzed GAM and ANOVA with only considering type and depth.
- ➔ Box plots are adopted in Fig. 2 instead of chart plots, to minimize lack of replication.
- ➔ The aim of the study (i.e OC changes with chronosequence) remained robust and intact, and after addressing the issue of limited sample sizes, conclusiveness has not deviated.
- ➔ English is rechecked by our expert member of team.

We thank you again, and looking forward to receive positive response after the first review.

**Rev#1 2nd round of comments to: bg-2021-359**

The authors have improved the quality of the manuscript since the last edition. There are still grammatical errors present throughout the manuscript. In the authors responses' to comments they stated someone has revised the English, I suggest this is done again as there are still mistakes in the English. Please change "stock" to "stocks: throughout. Also, the sample replicate number is still low. I have suggested one of the figures is changed give the readers a better understanding of the number of data points used and their spread.

Response: Thanks for the suggestion. We have checked language again and refined further. We thank our expert on editing, Dr. Charissa for that.

For the figure, as suggested, it has been changed to box plots for all the parameters.
Line16: I don't think mangroves can be considered 'macro-climate regulators'
Response: Agreed. This word is deleted from the main text.

Lines 90 to 163: I would include somewhere in one of these sections (study area or sampling procedure), a description of where in the sampling area samples were taken from. The parameters being measured in this study (mangrove OC etc.) have been shown to associate with distance to seaward edge or landward edge.

Response: Thanks for the comment. New sentence describing parameters is added under subsection 2.3

LN140-43 *The variables tested at each sampling sites from BS to MM were sediment thickness, coarse fraction, pH, ORP, bulk density, specific surface area (SSA), concentrations and isotope ratios of carbon and nitrogen, and pore water dissolved organic carbon (DOC).*

Description of study field has been provided in more details under a new subsection 2.2. Mangrove chronosequence

New sentences added as below-

LN 119-20 *Sediment sampling locations are different from each other in terms of mangrove development, elevation from mean sea level, and inundation pattern.*

LN122-30 *The ages of the mangroves are typically known from their plantation period (Salmo et al., 2014). In this study, mangrove categories are partly influenced by Fromard et al. (1998) who examined the chronosequential sedimentary OC in naturally growing Avicennia-dominated mangroves in the French Guiana muddy coast where PM were established on the seafront after stabilization of mud banks, or on the sandy offshore bar (height <2 m), followed by further maturation to younger stands (YM, height <8m). According to Fromard et al. (1998), both PM and YM colonize rather unstable marine clays/sands that are regularly flooded by tides. From the river mouth to upstream, the stands (adult and mature) become older and taller (8–15 m, Rhizophora spp. in French Guiana), phenomena that are linked to river dynamics rather than tidal movement.  In Bakhawan Ecopark, MM and AM sampling sites are farthest away from the water areas while BS and PM are closest to the sea (Fig. 1).*

LN132-34 *There is steep increment of elevation from seaward to landward sampling sites (-1.2 to 0.45m; refers to section 3). Seaward sites are featured by sandy sediment compared to silty clay sediment at the landward sites.*

Lines 107 to 108: What remotely sensed data? Is this a previous publication? If so, it should be referenced. If not, this data in supplementary information would be useful for readers.

Response: Thanks for the comment. We have now added citation for this statement.
LN107-08 *"Based on remotely-sensed data, it was found that the land area of the forest increased by 52.42% on average every five years since 1985 (Landicho et al., 2018)."*

Lines 117 to 119: I think it would be better for readers to give more detail about Marchand et al.'s (2003)

chrono-sequential design. It is unclear to me how certain ages of forest types were determined. How did the authors determine a 'young mangrove' stand? How did authors determine an 'adult mangrove'? Also, it seems quite circular to me that the purpose of this study is to look at OC accumulation/stocks in different mangrove ages, but the authors used a study that used OC to determine mangrove stand age.

Response: The main purpose of this study is to understand changes of sediment OC (stock and burial) and other associated parameters with development of mangroves of known ages (refers to section **2.2. Mangrove chrono-sequence and 4.1 Relevance of chrono-sequence approach).**

We have also replaced the Marchand e al., citation with Fromard et al., (1998) who first coined the terms like pioneer, young, adult, mature mangroves in French Guiana.

New text added:

LN122-23 *The ages of the mangroves are typically known from their plantation period (Salmo et al., 2014).*

LN123-30 *In this study, mangrove categories are partly influenced by Fromard et al. (1998) who examined the chronosequential sedimentary OC in naturally growing Avicennia-dominated mangroves in the French Guiana muddy coast where PM were established on the seafront after stabilization of mud banks, or on the sandy offshore bar (height <2 m), followed by further maturation to younger stands (YM, height <8m). According to Fromard et al. (1998), both PM and YM colonize rather unstable marine clays/sands that are regularly flooded by tides. From the river mouth to upstream, the stands (adult and mature) become older and taller (8–15 m, Rhizophora spp. in French Guiana), phenomena that are linked to river dynamics rather than tidal movement. In Bakhawan Ecopark, MM and AM sampling sites are farthest away from the water areas while BS and PM are closest to the sea (Fig. 1).*

Lines 416 to 493: Figure 1 shows, the MM site was furthest away from water areas, while BS and PM were closest. Which also corresponded to higher and lower C stocks in the present study. Authors should mention that mangrove carbon stocks have been shown to vary with distance to seaward and landward edges (references below) due to tidally driven nutrient cycling, OM retention and transport of allocthonous material.

Response: Thank you for the suggestion and citations. We have added them in the list and revised the text as

LN463-65 *At the Bakhawan Ecopark, OC stocks have been shown to vary with distance to seaward and landward edges (Kauffman et al., 2011; Wang et al., 2013; Chatting et al., 2020) due to tidally driven nutrient cycling, OM retention and transport of allochthonous material.*

Kauffman, J.B., Heider, C., Cole, T.G., Dwire, K.A. and Donato, D.C., 2011. Ecosystem carbon stocks of Micronesian mangrove forests. Wetlands, 31(2), pp.343-352.
Chatting, M., LeVay, L., Walton, M., Skov, M.W., Kennedy, H., Wilson, S. and Al-Maslamani, I., 2020. Mangrove carbon stocks and biomass partitioning in an extreme environment. Estuarine, Coastal and Shelf Science, 244, p.106940.
Wang, G., Guan, D., Peart, M.R., Chen, Y. and Peng, Y., 2013. Ecosystem carbon stocks of mangrove forest in Yingluo Bay, Guangdong Province of South China. Forest Ecology and Management, 310, pp.539-546.

Figure 2: Since the sample replicate number is low, I would recommend changing figure 2 to something other than bar charts. Possibly boxplots or even better would be faded boxplots with raw data points overlaid on top. This approach would make it clearer to the reader what the spread of the data was and would be more transparent regarding the number of data points these reported averages and confidence intervals are based on.

Response- Thank you for the recommendation on changing figure pattern.
In the revised boxplots, depth-wise data of each core were used reflecting sample size.

However, to clarify one thing that might help getting over confusion around sample replication/statistical analysis (GAM and ANOVA), kindly be noted that total 92 subsamples have been used to understand the effect of each explanatory parameter (type, depth) on $\delta^{13}C$, $\delta^{15}N$ and C:N ratio of SOM pool. We have 8 bulk cores in total and subsamples are 92. Sample sizes for each parameter are given in the Results section. Furthermore, we realized that our dataset is enough to evaluate the effect of age, and this is the most important point of this work. In contrast, we believe that the dataset is not sufficient to convincingly evaluate seasonal variations, and seasonal variation is out of the scope of chrono-sequential analysis. Hence, we reanalyzed GAM and ANOVA with only considering type and depth. Accordingly, Table 1, and text in Results and discussion are slightly modified. We did not change Fig 3,4,5 that is because to differentiate different collection period (i.e., dry and wet) from same collection sites (BS to MM).

[Figure]

*Fig. 2. Box plot diagrams of physical and biogeochemical parameters in the sediment subsamples according to mangrove development.*

**Reviewer#2**

The authors have improved the ms by re-arraning some parts of the text and by adding several smaller modifications. However, the main issue raised is not fully adressed: There is a mismatch between the sample size (5-6 cores) and the variables studied (age, season, depth). The uncertainty bounds in figure 8 are informative but do not provide an answer to this issue. I believe that the authors should be very clear about the exlporatory nature of their study and consider the limitations of their observational database in

both the interpretations and conclusions. A thoroughly revised ms that considers this could be considered for publication in bgs in my opinion

Response- Thank you for the constructive comment.

To clarify one thing that might help getting over confusion around statistical analysis (GAM and ANOVA), kindly be noted that total 92 subsamples have been used to understand the effect of each explanatory parameter (age, type, depth) on $\delta^{13}C$, $\delta^{15}N$ and C:N ratio of SOM pool. We have 8 cores in total and subsamples are 92. Sample sizes for each parameter are given in the Results section. Furthermore, we realized that our dataset is enough to evaluate the effect of age, and this is the most important point of this work. In contrast, we believe that the dataset is not sufficient to convincingly evaluate seasonal variations, and seasonal variation is out of the scope of chrono-sequential analysis. Hence, we reanalyzed GAM and ANOVA with only considering type and depth. Accordingly, Table 1, and text in Results and discussion are slightly modified. We did not change Fig 3,4,5 that is because to differentiate different collection period (i.e., dry and wet) from same collection sites (BS to MM).
However, we have acknowledged that overall number of cores are limited when considering age and stock relationship. In conclusion, we do not see any significant changes that might occur because of that. The rationale has been clarified in the discussion part (section 4.4 Increase of organic carbon with mangrove development

New text added –

LN228-29 *Results from subsamples of each core were used to for the model (total core = 8, subsamples used for analysis = 92)*

LN10-20 *Finally, it is important to highlight that the progression patterns of C stocks and/or CAR with mangrove age are observed out of total seven cores only, and the present dataset doesn't have enough numbers to test the effects of many other variables to the relationship between C stocks, CAR and mangrove age. Such relationship could be changed by environmental factors such as topography, hydrodynamics, geomorphology, biodiversity. We also found significant effect of soil depth on OC concentration (Table 1). However, as mentioned in section 4.1, environmental variabilities like hydrological processes do not largely vary among the sites, and have been relatively stable over ~ 30 years after restoration, and biomass development follows a similar trajectory of soil salinity, plantation spacing and species richness. Comparative data of chrono-sequential based OC accumulation rates with other restored mangrove forests also gives overlapping ranges, thus confidence of this work. Therefore, the results or conclusion of this study might not significantly change due to lack of replicates of sediment cores from the restores site. Nonetheless, acknowledging this as a limitation of the study, we further recommend that several cores are required for drawing a robust carbon and age relationships, especially for the regions where environmental variabilities can be significant drivers of these relationships.*

**New references**

Chatting, M., LeVay, L., Walton, M., Skov, M.W., Kennedy, H., Wilson, S., and Al-Maslamani, I.: Mangrove carbon stocks and biomass partitioning in an extreme environment. Estuar Coast Shelf Sc. 244, 106940, doi.org/10.1016/j.ecss.2020.106940, 2020

Fromard, F., Puig, H., Mougin, E., Marty, G., Betoulle, J.L., Cadamuro, L.: Structure, above-ground biomass and dynamics of mangrove ecosystems: new data from French Guiana. Oecologia 115: 39-53, doi.org/10.1007/s004420050489 1998

Kauffman, J., Heider, C., Cole, T.G., Dwire, K.A., and Donato, D.C.: Ecosystem carbon stocks of Micronesian mangrove forests. Wetlands. 31: 343-352. doi.org/10.1007/s13157-011-0148-9, 2011

Landicho, K.P., Blanco, A.C., Francisco, R.R., Gatdula, N.: Google earth engine-based assessment of expansion of Bakhawan Eco-Park using vegetation and water indices derived from LANDSAT images. Proceedings Asian Conference on Remote Sensing 2018. Page 332. ACRS 2018 PROCEEDING.pdf (a-a-r-s.org)

Wang, G., Guan, D., Peart, M.R., Chen, Y., and Peng, Y.: Ecosystem carbon stocks of mangrove forest in Yingluo Bay, Guangdong Province of South China. Forest Ecol Manag. 310, 539-546, doi.org/10.1016/j.foreco.2013.08.045, 2013.

---

## Author Response (AR3)

**Response to comments (third revision)**

**Comments to the author**:
Dear Dr. Ray:

Thank you for uploading your second revision which resolves the main issue of the referees (limited replication) through focusing on age rather than digging deeper into seasons, etc. I have read your paper in detail and there are still some issues pending, the majority technical/language related or sloppiness (multiple missing references).
I therefore invite you to revise again and submit a version that can, hopefully then, be forwarded to the production office.

Reply-

Dear Prof. Middelburg,

Thank you very much for your decision. We addressed all the comments in the present revised version. Your suggestive input are highly appreciated and we know they were super helpful for further refinement of the ms.
We look forward to your positive decision.

Best regards,
Raghab (on behalf of all co-authors)

**List of remarks:**

- Multiple references, Canfield, Bouillon et al. 2008, etc are missing. Please check carefully the completeness of your reference list.

Reply We have provided full refences of these two citations and carefully checked all that included in the ms.
- Page 1 title: shouldn't it be observationS? Corrected
- Line 80: observations Corrected
- Line 95: depositing sediment to form Corrected
- Line 108 (but all through): check the number of significant digits. An average of 52.42% does not make sense. Reply: Thank you for the suggestion. We have now reduced the decimals at significant level
- Line 155: carbon and nitrogen. (carbon stocks are estimated not measured). Corrected
- Line 182: give the G-force, not only 2000 rpm
Reply G-force is added in the ms as 760 g for the rotor we used in analytical purpose.
- Line 241: increase of the number of end-members for the model Corrected
- Line 256: I guess you should have written: showed higher values, otherwise I do not understand the sentence (discrepancy with figure). Corrected
- Line 298: Replace on the other hand with However, because on the one and on the other always come together Reply- Thank you for the correction.

- Line 365: is Ray et al. already published by now. And indicate whether negative means efflux or influx.

Reply - Since it is not published yet, we have rephrased it simply as 'unpublished data'. Signs of influx and efflux are given now.

- Line 374-375: logic of sentence needs attention. .. plants,…., as it is composed…

Reply – Corrected as Terrestrial C3 plants, like mangrove plant organs, have C/N ratios of around 12 or higher (Prahl et al., 1980) and are N-poor due to the dominance of lignin and cellulose type of compounds.

- Line 391: -21.07 permille. This number of digits does not make sense given the lumping of benthic and pelagic algae and their spread (Fig. 6).

Reply: Thank you for the suggestion. We have now reduced the decimals at significant level

- Line 424: Porewater profiles…..sediments are very rare…

Reply- Corrected as "Porewater profiles of salinity and DOC in mangrove sediments are very rare.."

- Line 436: delete Ray et al. unpublished here. No need and do cite Bouillon et al. 2008 in reference list.

Reply- Deleted and full citation of Bouillon et al is given in the list.

- Line 499: adult/mature.. Corrected

- Line 500: reformulate ".. essentially not very significant ''. Simply not significant, or not?

Corrected as not significant

- Line 501: … might be constrained by some biological or geophysical factors. (delete significantly, if significant do not use might be). Deleted significantly

- Line 561: down in the tidal flat Corrected

- Reference list: please check careful before resubmission.

Reply- References are carefully checked and formatted.

**Newly added references in the list**

Bouillon, S., Borges, A.V., Castañeda-Moya, E., Diele, K., Dittmar, T., Duke, N.C., Kristensen, E., Lee, S.Y., Marchand, C., Middelburg, J.J., Rivera-Monray, V.H., Smith III, T.J., Twilley, R.R.: Mangrove production and carbon sinks: a revision of global budget estimates. Glob. Biogeochem. Cy. 22, GB2013. 2008, doi.org/10.1029/2007GB003052,2008.

Canfield, D.E.: Factors influencing organic carbon preservation in marine sediments. Chemical Geology, 114, 315-329, 1994. doi.org/10.1016/0009-2541(94)90061-2

---

## Author Response (AR4)

**Response to comments (third revision)**

**Comments to the author**:

Dear Dr. Ray:

I have read your revised version and I am happy to inform you that your paper is almost ready for publication. A few technical corrections are needed though. I list these below.

With best regards,

Jack

Response:

Dear Prof. Middelburg,

We thank you very much for supporting our work throughout the review journey.
We honestly feel that after series of corrections on those constrictive comments from you and the reviewers has made the manuscript scientifically robust, meaningful, and surely a well-read iteration.
In this version, we have accepted all of your edits with few minor restructuring of sentences.

Thank you once again for accepting this work.

With best regards
Raghab (on behalf of co-authors)

Line 41: …ecosystems that have the potential to climate change mitigation…
Corrected
Line 69: Most of the …. Sediments changes…. Corrected
Line 140: ORP is used without explanation. Introduce the full term here Full name is given now
Line 273: …showed wider variations… Corrected
Line 384: allochthonous refers to material from elsewhere, but benthic algae are locally produced. Or do you specifically mean resuspended benthic algea.

Edited as
Similar values of $\delta^{13}$C in the surface water POC upstream (–25.9‰, Table S2) and sediment OC at the bare sediments and pioneering mangroves (mean –26‰) might reflect resuspended benthic algae as major sources of SOM.

Line 407-409: The logic of this sentence needs attention: hypoxic or anoxic conditions I presume and I do not see how ..and creating connect with the first part of the sentence.

Edited as

In Kenya, greatest OC concentration was measured where the crab population was maximum, particularly at low tide when the presence of water with low oxygen saturation covered the bottom of the burrows that avoided oxidation and created an extension of the coastal marshes sediment-air interface favoring greater OC (Kristensen, 2004; Smith III et al., 1991).

Line 473: greater biomass and sediment TOC, porewater DOC….Corrected
Line 493: pelagic continental shelf sediments do not make sense. Delete pelagic deleted